# Statistical Study of User Perception of Smart Homes during Vital Signal Monitoring with an Energy-Saving Algorithm

**DOI:** 10.3390/ijerph19169966

**Published:** 2022-08-12

**Authors:** Carolina Del-Valle-Soto, Juan Arturo Nolazco-Flores, Jose Alberto Del Puerto-Flores, Ramiro Velázquez, Leonardo J. Valdivia, Julio Rosas-Caro, Paolo Visconti

**Affiliations:** 1Facultad de Ingeniería, Universidad Panamericana, Álvaro del Portillo 49, Zapopan 45010, Mexico; 2School of Engineering and Science, Tecnológico de Monterrey, Monterrey 64849, Mexico; 3Facultad de Ingeniería, Universidad Panamericana, Josemaría Escrivá de Balaguer 101, Aguascalientes 20290, Mexico; 4Department of Innovation Engineering, University of Salento, 73100 Lecce, Italy

**Keywords:** elderly care, energy saving, robotics for healthcare, smart home, vital signal monitoring, wireless sensor networks

## Abstract

Sensor networks are deployed in people’s homes to make life easier and more comfortable and secure. They might represent an interesting approach for elderly care as well. This work highlights the benefits of a sensor network implemented in the homes of a group of users between 55 and 75 years old, which encompasses a simple home energy optimization algorithm based on user behavior. We analyze variables related to vital signs to establish users’ comfort and tranquility thresholds. We statistically study the perception of security that users exhibit, differentiating between men and women, examining how it affects the person’s development at home, as well as the reactivity of the sensor algorithm, to optimize its performance. The proposed algorithm is analyzed under certain performance metrics, showing an improvement of 15% over a sensor network under the same conditions. We look at and quantify the usefulness of accurate alerts on each sensor and how it reflects in the users’ perceptions (for men and women separately). This study analyzes a simple, low-cost, and easy-to-implement home-based sensor network optimized with an adaptive energy optimization algorithm to improve the lives of older adults, which is capable of sending alerts of possible accidents or intruders with the highest efficiency.

## 1. Introduction

Technological advances, which continuously emerge to satisfy human needs, help us delegate tasks and minimize costs, workload, and time. Technology changes people’s lifestyles and even creates new needs with innovative products. The creation of multiple systems that automate processes requires a centralized control, which saves resources, and constant monitoring, which allows us to obtain information on its functionality [1]. Currently, intelligent design applied to the home sector has become an interesting approach to endow such spaces with safety and functional technology [2]. For intelligent design, we refer to home automation that enables interactivity between the space and user. It is an associated system of sensors responsible for automating and monitoring various aspects of the home environment, such as energy consumption and savings, as well as domestic security, and can assist in the prevention of accidents [3]. The ultimate aim is generate safety and comfort.

People have an ever-increasing life expectancy. Older adults want to be as independent as possible. However, the lack of elderly population monitoring is a major problem, especially for those lacking consistent attention from family members, caregivers, or a health center. The situation can get complicated, even leading to the death of the person [4]. Older people acquire certain routines in their daily lives, which allows a sensor network to “learn” some activities depending on the specific day or time. In this way, the network is capable of managing the behavioral patterns observed in the user, optimizing their correct interpretation through the information obtained by the sensors installed at home [5]. Undoubtedly, sensors play a major role in the prompt detection of behaviors related to motor or cognitive problems.

Internet access has become a primary need for human development because multiple activities currently require a connection to the web. Education and work are the essential activities that use this service, and they constitute fundamental pillars in the economic development of our society. Highly performant solutions have been generated using WiFi standards and mesh communication technologies [6]. The current global crisis caused by the COVID-19 pandemic has imminently hastened the virtualization of activities and, therefore, the use of Internet services [7]. Older adults are currently at greater risk of blackouts or accidents at home [8]. The World Health Organization (WHO) reports that falls are the second leading cause of death from accidental or unintentional injuries. Those over 65 exhibit the highest prevalence of fatal falls [9].

The technology necessary to remotely detect behavioral changes in real-time is based on recognizing activities from the information gathered from sensors. Public health services are facing shrinking budgets and an increasing pressure to cut costs. Considering the scarcity of specialized healthcare places for the elderly, the solution is to keep them at their own home [10]. For the elderly, representing a significant part of social expenses, it means the obligation to live alone and independently at their homes, with all the risks that this entails. To respond to these new needs, research has been paramount for the meaningful development of telemedicine systems over the last 20 years. These systems are designed to offer greater security to people that live alone at home, including those in specialized centers, and represent a valuable tool for caregivers [11].

Information and Communication Technologies (ICT) have become essential for promoting independent living and improving the quality of life of the elderly. These people experience a progressive loss of functions due to their age or chronic diseases, which makes it difficult for them to carry out daily tasks, forcing them to depend on third parties or, at least, periodically inform someone else about their health status. Digital home is a new technological domain deployed in places of residence to increase security and improve comfort and the ease of communications. All this is achieved with the use of telecommunications networks, the integration of services, and the interconnection of equipment and facilities [12].

Our work is framed in this context. In this study, we analyze the impacts of a sensor network on the life of an older adult at home [13], optimizing its performance in favor of energy savings. The originality of our study lies in a proposed algorithm for a simple sensor network for home care. This algorithm is characterized by its low energy performance for this type of application. It bases its low consumption on the hierarchy and clustering of the sensors according to a person’s habits. We study the person’s well-being based on four variables expressed by their vital signs: heart and respiratory rates, body temperature, and sleep rhythm. This behavior is studied to analyze its impact on the main areas at home. We analyze the behavior of these variables for two weeks. In the first week, the sensor network monitors the person at home. In the second week, the person is at home without network monitoring.

In addition, we perform a statistical analysis on the impact of the person’s sense of well-being with the network, considering men and women independently. Through this gender analysis, we observe the algorithm’s behavior concerning the energy consumption of the sensors in houses containing female, male, or mixed occupants. Finally, we establish the degree of usefulness of the sensor network in the home to observe the alerts of the devices in the event of possible accidents or anomalies in the installation conditions, and we analyze the effects of the algorithm’s response under safety conditions. Therefore, this work addresses two important objectives: (a) to statistically analyze the perceptions of safety and comfort by older adults using the monitoring network for their care; (b) to adapt the sensor network to low energy consumption through an algorithm that reduces the cost of the home’s electrical energy while maintaining the quality of life of the elderly.

### 1.1. Related Work

Currently, a growing number of older adults live alone in their usual residences, despite the risks that this entails [14]. Aging inevitably results in diminished sensory acuity, decreased muscular endurance and strength, impaired mobility, decreased mental clarity, and altered stability. Technologies included in smart homes can be adapted to the needs of the elderly in order to improve their quality of life. Despite the drawbacks previously noted, a significant number of older adults actually prefer to live on their own [15]. A medical alert system’s automatic fall detection feature provides peace of mind that the person will receive the care they need. Fall detection devices use alert system technology to identify and provide emergency assistance. These devices can be placed on a person’s neck, wrist, or waist, or can be used as fall detection devices that can be worn separately with a medical alert button [16]. These solutions are simple and easy to implement but require the person to wear an accessory at all times; otherwise, they would be of no use. This is where a system that does not require constant attention is a perfect option.

Familiarity with the area of residence, being able to manipulate the environment according to needs and preferences, the feeling of autonomy and independence, and the security provided by a family environment are clear indicators that older adults try to extend their stay as much as possible in those places where they have lived for many years [17]. This concept is called “aging in place” and involves a comfortable environment, a feeling of familiarity and safety, and a neighborhood with familiar people to interact with, creating a certain sense of autonomy and independence. The convergence between the needs of the elderly and the currently existing technological solutions have been previously studied in some works. The satisfaction of these needs improves their quality of life, promoting an autonomy that, inevitably, is reduced over the years. There is an indisputable desire from the elderly to remain at their usual places of residence. In this sense, home automation can significantly ensure that older people continue to reside at their own homes, adapting such spaces to their specific needs [18,19,20].

The increasing attention to energy efficiency in housing stimulates the expansion of smart homes. Naturally, a balance is required between energy efficiency and the needs of the occupants. The work cited in reference [21] presents the challenges faced when exploiting non-invasive wireless devices for user-behavior monitoring. Two application fields are addressed: intelligent energy management and elderly monitoring. This work contrasts well with our approach from the point of view of the occupation of the person detected non-invasively and energy savings in detection systems. As in our work, the authors explore real cases in homes to analyze the network. Another related study, although from a more general perspective, is the one cited in reference [22]. The authors discuss the ecological implications of smart meters, devices, and homes. Considering their approach, our work meets the Home Energy Management Systems requirements, such as minimizing energy demand, costs, security, and flexibility, as well as risk management. Likewise, the work reported in reference [23] presents an international selection of the main smart home projects and the associated technologies of wearable/implantable monitoring systems and assistive robotics. This work reviews functions that can be implemented in smart homes and the related equipment, objects, and appliances. In this way, it complements our study by focusing our objective on classifying smart homes or systems with their used equipment, fundamental algorithms, and functions.

It is also worth mentioning the studies exploiting wearable sensors. In reference [24], the authors comprehensively analyze the concept of telemedicine by focusing on the client and server sides. They show that other studies associated with IoT-based smart home applications have several limitations that have been not yet addressed, such as remote patient monitoring in home healthcare applications using IoT. In other literature review studies, such as reference [25], the authors present a systematic review of the literature to determine the technology readiness levels (TRL) of the elderly, as well as exploring the effectiveness of smart-home-based information-monitoring technologies for older adults. This last approach addresses an important aspect considered in our work. Thus, one of our objectives is to approximate the prediction of disability- and health-related qualities of life non-invasively through a sensor network. This problem can be extrapolated to remote medical care. For example, in reference [26], healthcare staff can also track the general health status of older people in real-time, providing feedback and supporting them from distant facilities. This paper presents a comprehensive review of cutting-edge research and development in smart-home-based remote healthcare technologies. This work is complemented by reference [27], in which the authors provide an overview of the current state of intelligent health-monitoring systems. They review these systems in intelligent environments from a general perspective and with particular attention paid to systems for the elderly and dependents. They also discuss the challenges for these systems from the perspective of technology development, system requirements and design, and modeling.

### 1.2. The Needs of the Elderly

This work intends to contribute to improving people’s quality of life. The notion of “quality of life” includes a wide variety of concepts, such as “satisfactory life”, “decent life”, “subjective well-being”, “psychological well-being”, and “personal development”. Factors that reduce quality of life are diverse: decreased physical abilities; psychological factors, such as personality traits, loneliness, and feelings of worthlessness; lack of security, inadequate living environment, isolation, social/economic deprivation, etc. [28]. Improving a person’s quality of life is carried out using a non-invasive sensor network, which has an algorithm for optimizing energy in the home. The main stumbling block is that many of our elders are reluctant to change and unfamiliar with new technologies. For this reason, there are proposals related to non-invasive technologies for the care of people [29], although they are still wearables. However, what is intended with innovation in care for the elderly is not only to create tools for personal use but also to facilitate the day-to-day life of caregivers, optimizing their time and work.

Helping the elderly stay healthy and ensuring a good quality of life in their later years is one of the biggest challenges for today’s healthcare sector. For that reason, independence and autonomy, care for the person’s functional capacities, and dependency prevention must be promoted [30]. The progressive deterioration of the elderly creates an urgent need: attention to their health status. The objective of measures and services for the elderly is to avoid this deterioration as much as possible.

### 1.3. Technologies for the Elderly

The benefits of the implementation of technologies in the places of residence of the elderly must be measured in terms of the extent to which older people will be empowered to lead an independent life, with these technologies preventing their social exclusion by providing alternative ways to communicate. Some advantages are optimizing private and personal life and increasing personal security [31]. These systems provide constant security to individuals, thanks to the active monitoring of their activity and reducing dependency by enabling the older adult to perform functions he/she previously required help with. Remote management (using electronic devices) simplifies home management and automates tasks that the person cannot otherwise perform (or has difficulty doing) on their own [32].

Some limitations of these technologies are their high cost and handling complexity. The latter is a major barrier because the elderly are usually reluctant to use new technologies. Another important aspect is the person’s feelings of being “observed” or “watched” constantly [33]. This leads to feelings of intimacy/privacy invasion and, in this way, the rejection of healthcare technology [34].

### 1.4. Requirements of a Solution Based on Technologies for the Home

The main characteristics of any solution that seeks to alleviate the needs of the elderly and achieve a high degree of acceptance are framed in the economy, usability, adaptability, and functionality of the proposals. Solutions must be economically affordable. Systems must be easy to install, maintain, and use. They must be flexible and scalable, so that future service expansion or reduction is transparent [24]. Their functionality must be well-known and adapted to the existing needs. Furthermore, a key aspect is that these devices must be customizable to each user.

Domotic systems allow us to solve the needs of the elderly in an isolated way or through integrating joint solutions that manage to include all the individual solutions. Currently, the most popular technological solution implemented in homes is the remote alarm service, which allows people to activate an alarm in the event of an accident [35]. This system has evolved into a complete telecare service. In addition to managing the alarm, it offers a social support service to older adults. Most service providers offer the management of other types of home help as well, such as repairs or maintenance. Proposed technologies allow the automation and coordination of all the devices at home that can be controlled to simplify and improve the quality of life of its residents [36]. It is all about integrating device management, programming actions, and interacting in a friendly way with the system, so that the environment can be easily controlled [37].

Home automation provides comfort, communication, and security, and efficiently manages energy use, ensuring water, electricity, and fuel savings [38]. The demand in the market continues to grow, and the functions of each home are changing. Wireless sensor networks for the home aim to reduce global energy consumption and CO${}_{2}$. Theoretically, by integrating a home automation system, up to 60% of electricity consumption can be saved [39]. In addition to the need to reduce energy consumption to decrease pollution, the factor must be considered economical.

## 2. Materials and Methods

With the rapid increase in real-time human health monitoring and the seamless interaction between humans and machines, many research efforts have been made in recent years on wearable sensors. Figure 1 exemplifies different types of sensors to interact with human beings. Some of these sensors perform real-time measurements of body temperature, heart rate, pulse oxygenation, respiratory rate, blood pressure and glucose, electrocardiogram, electromyogram, and electroencephalogram signals, and so on.

In this work, we implemented a network with different sensors, such as level, pressure/temperature/humidity (PTH), motion, noise, gyroscope, light, and air, to monitor the home’s main spaces. We presented a descriptive experimental investigation of an average house in Guadalajara, Mexico. We invited forty test subjects (20 women and 20 men) aged between 55 and 75 years old to participate in our research. We deployed the network at each subject’s home for one week and removed it for the second week. In both weeks, we measured four vital signs of the subjects: heart and respiratory rates, body temperature, and sleep rhythm.

The implemented sensors encompass the work of an intelligent system by analyzing the different stimuli at home, transforming them into a digital impulse captured by a more robust device called a collector node. We carried out the experiments in 40 residences in the city of Guadalajara, Mexico, for one week. The measured body variables are pertinent for the subjects’ daily life at home. The internal and external house temperature helps regulate the indoor climate and predict the way of dressing. Presence monitoring in the various sectors of the house helps quantify the passage of people in each one of them. Light-intensity detection controls the lights turning on and off, according to the time and the day’s conditions. Moisture is one of the leading causes of deterioration in home living, and on many occasions, it is invisible to the eye. Noise sensors are capable of picking up a considerable amount of sounds, not just voices.

### 2.1. Variables

We consider the variables as follows: First, heart rate is the number of times the heart beats per minute (bpm). Measuring heart rate is a clear indicator of health. In addition, it allows us to know whether a person is exercising correctly. The standard heart rate at rest is 60 to 100 bpm. Second, respiratory rate is the number of exhalations a person takes each minute. It is one of the principal vital signs together with blood pressure, pulse, and temperature. The respiratory rate changes according to many health and activity factors. The regular respiratory rate for healthy adults is 12 to 20 breaths per minute. Carbon dioxide leaves the lungs at the same rate that the body produces it at this breathing rate. Respiratory rates below 12 or above 20 may imply a disruption in normal respiratory processes. Third, body temperature measures the body’s ability to generate and remove heat. There are four ways to measure temperature, namely under the armpit, in the mouth, in the ear, and in the rectum/vagus. Not all people have the same “normal” body temperature. For a typical adult, body temperature can range from 36.1 ${}^{\circ }$C (97 ${}^{\circ }$F) to 37.2 ${}^{\circ }$C (99 ${}^{\circ }$F). The body temperature does not remain constant through the day and varies throughout life. Finally, sleep rhythm is measured using a variable called actigraphy. Actigraphy records movements through a sensitive accelerometer, typically worn on the wrist, and estimates sleep parameters using a computerized algorithm. It is a valuable test in adults, simple to perform, and increasingly available to the general population. The overall sleep score on a smartwatch is a sum of the individual scores for sleep duration, quality, and recovery, with a maximum total score of 100. Most people score between 72 and 83. The sleep score ranges are as follows: Excellent: 90–100, Good: 80–89, Fair: 60–79, Poor: Less than 60. These scores consider metrics such as time asleep and awake, how much a person sleeps, and the quality and recovery (turns that decrease quality) of sleep. Most smartwatches and fitness trackers have a three-axis accelerometer and gyroscope to measure orientation and rotation. Through actigraphy, the smartwatch translates wrist movements into sleep patterns. Yet, these activity bracelets or smartwatches cannot accurately discriminate the stages of sleep.

### 2.2. System

Sensors used in home automation are typically small devices capable of detecting and reacting to the different changes that occur around them. The proposed system is shown in Figure 2. This network presents the circuits in a plastic box with a pleasant visual presentation in order not to be invasive with the design of the house, as well as to avoid electronic circuits having wires exposed to damage or accidents. The collector device is a radio frequency node operating in the 2.4 GHz band with the IEEE 802.15.4 communications protocol. Working as a sensor node, this device has a sensor device for physical variables, such as pressure, temperature, and humidity sensors, infrared presence and sound level sensors, actuators, hydrogen potential, dissolved oxygen, etc. This device is responsible for sending the information from the sensors to the network and to transmit such information to the concentrator radio node associated with the computer. Data are processed in a convenient way for the user or the developed system for decision making. This device can also act or function as a repeater node for the data obtained by a sensor node. It retransmits the data between different repeaters or sensor nodes until it reaches the hub node associated to the computer. The communication protocol with the self-organized wireless network is responsible for telematics, telemetry, and radio frequency. However, it is easy to implement in the user’s computer or even to develop its software because it is open source. On the other hand, the access control to the medium and the physical layer of the radio frequency nodes comply with the IEEE 802.15.4 standard.

The distributed sensor network consists of 14 nodes. There is a concentrator or coordinator node, which is the device connected to the computer through a USB connection and receives/manages all the information from the network. In addition, the architecture consists of 13 router/sensor nodes. These devices can have sensors detecting temperature, pressure, humidity, infrared presence, light, sound, etc., connected to their communication ports. The sensed parameters are transmitted via radio frequencies to the concentrator or coordinator node in the network. They can also act as signal repeaters without sending additional information to the network or parameters of connected devices because they only retransmit the packages that other nodes are generating.

The radio frequency nodes have a port- or connector-type SMA (sub-miniature for high frequency) for the connection with the RF antenna, a double communication port via RS485 protocol, as well as a USB communication port.

Figure 3 shows the node circuits used for the router node and for the radio frequency nodes. This schematic describes the inputs and outputs of the sensor circuit boards, the connection ports, and the antennas. In addition, the voltages and currents allowed for each type of power supply are shown. The sensor’s elements, such as capacitors, resistors, and integrated circuits, with their respective polarizations, can be appreciated. Finally, the frequency at which the sensors work is displayed.

The implemented sensors target the main activities that people conduct at home. They are related to the house sectors where a person commonly interacts with other people or with the infrastructure.

### 2.3. Sensor Specifications

**(A) Pressure, Temperature, and Humidity (PTH) Sensor.** Operating voltage: 1.8–3.3 V DC; pressure range: 300 to 1100 hPa (0.3–1.1 bar); temperature range: −40 ${}^{\circ }$C to 85 ${}^{\circ }$C; temperature resolution: 0.01 ${}^{\circ }$C; temperature accuracy: 1 ${}^{\circ }$C; relative humidity range: 0–100% RH; relative humidity accuracy: +−3%; ultra-low power consumption; sampling frequency: 157 Hz (max.).

**(B) Motion sensor.** These sensors are capable of detecting when a person approaches a specific area. They are placed at different strategic points of the house to know if the user has carried out all the actions that he/she is supposed to do during his/her daily routine. Upon the information gathered by the sensors, the system can create a table of activities per day, week, and month.

**(C) Gyroscope sensor.** These are placed on the doors at the user’s home. The functionality of these sensors is to detect when the user has received a visit and how long the visitor stayed. It is also possible to visualize when the user has left home and how long the house remained unattended.

**(D) Noise sensor.** Input variable: DC 3.3–6 V; sensitivity: adjustable; digital output when detecting sound.

**(E) Level sensor.** It is an electronic device that measures the height of any object. Generally, this type of sensor works as an alarm that is triggered when the threshold level has been reached.

**(F) Light sensor.** It is a device with many applications. It is especially useful to make a much more efficient use of the energy at home, adapting the luminaries’ power to the existing ambient light.

**(G) Air quality sensor.** It allows for measuring the quality of the air. This sensor warns of high concentrations of CO${}_{2}$ gases in the air, which affects people’s health.

### 2.4. Algorithm

The proposed algorithm adapts the sensor network to the person’s living conditions. This adaptability operation saves energy. The nodes present adaptive hierarchies learned based on the person’s way of life. The sensors adapt their hierarchy IDs to prioritize those uses of the house. These hierarchies are learned over three days, and then the sensors adapt the network topology so that devices with higher hierarchies have more continuous measurements of the person or place in the house. Additionally, the algorithm has a time of 4 h in which if it does not detect activity, the sensors enter a sleeping mode. This period is not taken into account if it is during the night. This sleeping mode saves energy in the nodes when people leave the house or are traveling.

Algorithm 1 details the process of optimizing energy in the network, basing the savings on the routing protocol’s operation and the nodes’ hierarchies. The value is the parameter that each sensor measures. This proposed algorithm saves energy in network control packets and, therefore, in the network overheads. This is because the formed groups (clusters) change their management of the routing protocol to a reactive or proactive nature depending on the node activity. Sensors have a higher hierarchy when their change threshold exceeds a specific steady-state value. This means that the sensor has more activity or is used more in that area than others. In each zone of the house, there is at least one motion sensor. Therefore the algorithm can detect the most continuous presence in each zone of the person’s home. In addition to hierarchies, clusters of nodes are created to share energy costs. Clustering promotes topology maintenance and efficient power routing by transferring a significant portion of the communication overhead to the cluster head. The nodes with a higher hierarchy create a cluster, and with this, the routing protocol acts reactively to keep the nodes more alert. The lower hierarchy clusters are kept in sleep mode, and the routing protocol changes to a proactive nature to have more spaced control-packet exchange times.
**Algorithm 1** Pseudocode of the algorithm.StartRequire: coordinator node starts;
 Per each node do:Set Hierarchy = 0;Set allowed_value;Set day = 1;while time <= end_time do:  Set avg_met_vector[] <-- average met per day;  Set std_dev_met <-- standard deviation met;  if day > 3  for i = 0; i < length(avg_ind_vector[]); i++     avg_met <-- average metric value;     if avg_met >= allowed_value        Hierarchy++;     end if  end for  end if  day++;end whileend doNodes with same Hierarchy form a cluster;per each ID_cluster do:   if ID_cluster_i > ID_cluster_j      Set nodes from ID_cluster_i in active mode;      Set nodes from ID_cluster_i with reactive      routing protocol;      Set nodes from ID_cluster_j in sleeping mode;      Set nodes from ID_cluster_i with proactive      routing protocol;   end ifend doend

Figure 4 shows the general scheme of the operation of the proposed algorithm. Initially, the nodes are distributed throughout the house, and the person moves freely within it. All sensors require three days to obtain an average value of their measurement metric. When the person spends more time in the same place in the house, the sensors in that area increase their hierarchy and create a cluster. The other sensors remain in their current hierarchy. Sensors with a higher hierarchy are put into active mode and their routing protocol works reactively. The other sensors are put into sleep mode for specific periods and the routing protocol works in proactive mode.

### 2.5. Baseline Conditions of the Older Adults for Experimentation

We included a simple database that asks each person to be considered as part of the sample. We included the initial experimentation questionnaire for classifying people and selected healthy people without heart or respiratory conditions. We added a table to clarify the initial conditions of the investigation to be able to assess, as far as possible, our experiment correctly. We chose healthy non-smoking people that practice sports at least three times a week. Next, we presented the set of measured characteristics and questions that were asked to older adults to be considered in this work.

## 3. Results

Usability tests were conducted with the 40 test subjects to assess the system’s performance. This section presents the tests conducted over two weeks with older adults living in an average home in Guadalajara, Mexico. For the first week, the measurement of the four vital sign variables (heart and respiratory rate, body temperature, and sleep rhythm) is proposed with the sensor network (we will call this scenario “with sensors”). For the second week, we continue measuring the vital signs but the network is no longer installed at the user’s home (we will call this scenario “without sensors”). The network consists of the set of sensors described above. Sensors are connected to the 14 distributed nodes along the house (see Table 1 and Table 2). This network monitors the basic activities of the user at home. Not all the sensors work to generate healthcare alerts. Some of them are just meant to ease the living conditions and improve the quality of the user’s daily activities. For example, luminosity sensors are used to control luminaries by setting them on/off depending on the time of day, so that the light intensity changes automatically according to the user’s needs. Gas sensors are used to detect possible gas leaks, thus preventing gas intoxication and risks of explosion. Sensor gyroscopes are placed on windows and doors to record their opening and closing. Infrared (IR)-type level sensors involve an IR emitter and receiver that form an invisible barrier; when an interference occurs, the IR ray interruption will activate the alarm device. Their main advantage is that they are fast/easy to install and difficult to be removed. For measuring pressure, temperature, and humidity, the different systems use thermostats, temperature sensors, or probes, the electrical output signals of which are proportional to the real temperature values. This sensor measures atmospheric temperature, barometric pressure, and relative humidity. Finally, the motion sensor detects and reacts to physical movements. It can be used for security and even for energy consumption control.

Table 2 and Table 3 show the general sensor distributions in all the house rooms, including the outside. Table 2 shows the nodes that comprise each house zone. These nodes can have one or more sensors connected to them. Then, Table 3 shows each type of sensor’s ID concerning their function and measurement parameters. The distributed sensor network consists of a concentrator or coordinator node, which is the device connected to the computer via USB connection and receives and manages all the information received or transmitted from the network. In addition, it consists of three router/sensor nodes (7, 10, and 14). These devices have sensors connected to their communication ports, such as temperature, pressure, humidity, infrared presence, light, sound sensors, etc. The sensed parameters are sent via radio frequency to the network to the concentrator or coordinator node (sensor 1). However, they can also act as signal repeaters without sending additional information to the network.

Figure 5 shows the distribution of the nodes in each of the house sectors. These devices can have one or more sensors connected for a complete monitoring tailored to each housing area. Figure 5 shows the average layout of a house between 150 and 250 m${}^{2}$.

Figure 6 shows the average occupation of the main areas of an average house. As previously mentioned, we consider a sample of 40 people. We took occupancy samples from the main areas of an average house for seven days. We found that subjects spend around 48% of their time in their bedrooms, followed by the kitchen, which has around 20% occupancy. Subsequently, the living room, the dining room, and the bathroom exhibit a similar occupancy, at around 9%. Finally, people spend less time in the garden—around 4%. This information gives us an idea of the ideal distribution of the sensors at home for the efficient care of the elderly.

Table 4 shows the change in the heart rate variable when people have an active sensor network. This variable is measured in each area of the house. The table shows that such a variable decreases in the bathroom, main bedroom, and kitchen, respectively. This may be due to the peace of mind that the older adult experiences when she knows that she has the monitoring of specific parameters that make her life easier and more comfortable.

Table 5 reflects a search for works related to ours and shows a series of main analysis characteristics for this type of experiment. We report the type of mechanism used for caring for older adults at home. In addition to this mechanism, the wireless or wired system is proposed to track or locate the person. In addition, we report if there is an energy-saving technique to optimize network and home resources. Additionally, we observe the topology of the network of technological resources and if this care has a positive perception in people. Finally, we report the tests’ execution times to validate the results.

## 4. Discussion

The performance of the proposed algorithm is initially tested concerning regular network operation for one week. We evaluated the energy consumption of the network in each sensor based on the remaining energy (RE) of the batteries using a Fluke 115 3.75-digit (6000) LCD digital multimeter. Table 6 shows the performance of the remaining energy in each sensor in the home monitoring network with and without the application of the algorithm.

Table 6 shows the energy savings using the proposed algorithm, with an average of 10% lower consumption in the sensor nodes. This saving of power consumption uses the nature of the routing protocol. It gives hierarchies to the nodes to deactivate them with less utility for periods. The novelty of this algorithm is that it adapts to the life of the elderly, not the other way around.

### 4.1. Degree of Confidence of the Sample within the Period of Experimentation

In order to show one week is enough, we calculate 95%, and then we calculate the maximum likelihood estimator. Figure 7 shows the one-week sampling of the HR of three persons. The visual statistical analysis (Figure 8 and Figure 9) shows that heart rate distribution is highly Gaussian. We observe that the histogram, Figure 8, follows a Gaussian distribution. The Q–Q plot, Figure 9, also shows the histogram is a high normal distribution, with very few outliers. Moreover, statistical analysis using the Shapiro–Wilk test fails to reject the hypothesis that the histogram is non-Gaussian.

The 95% confidence intervals of the heart rate for these three persons are shown in Table 7.

Similar behavior is observed for the other persons and measurements.

The maximum likelihood estimator is the sample mean: $X_{M}LE=\left(1/N\right)SUM\left(x_{i}\right)$, where SUM is over all samples. Additionally, $x_{i}$ is the sample data set of length *N*. Table 8 shows the mean estimators for each of these three persons.

Similar behavior is observed for the other persons and measurements.

### 4.2. Statistic Analysis

We are particularly interested in determining whether the subjects’ data suffer significant changes when subjects are aware that they are being observed by cameras. To validate this hypothesis, for each subject, we also captured data prior to installing the cameras. Since the observed measurements were captured under two different conditions (with and without cameras), we obtained a pair of observations for each subject. We want to verify if the observations collected under these two different conditions have an effect on the subjects. This is achieved using the “paired sample test”, also called the “dependent sample test”. If the observations are normally distributed or if the number of samples is “large enough”, we can use the *t*-test, otherwise, we have to use a non-parametric approach. In statistics, non-parametric tests are statistical analysis methods that do not require a distribution that meets the required assumptions to be analyzed (especially if the data are not normally distributed). Due to this reason, they are sometimes referred to as distribution-free tests. Non-parametric tests serve as an alternative to parametric tests, such as *t*-test or ANOVA, which can be employed only if the underlying data satisfy certain criteria and assumptions—the Kruskal–Wallis test.

(A) Testing Normality:

By plotting the box plot for categories, we visualize associations between these two categories. Figure 10 shows these box plots that allow us to visualize the type of association between “sensor” and “no-sensor” categories. Visually, the statistic association between the monitored values for “Breathing frequency”, “Heart rate” and “Body temperature” for all subjects before and after installing sensors are positive. Not one of them shows a much higher association than the others. On the other hand, visually, the statistic association between the monitored values for “Sleep rhythm” for all subjects before and after installing sensors are strongly negative.

The first step in this analysis is to test if data are normally distributed. We use a histogram to see if the sampled data follow a normal distribution; we also draw a box plot to see how well balanced are each of the quartiles. Figure 11 shows the histogram of the difference of the different measurements taken; in red we also draw a red line with the sample mean. From this figure, it is difficult to observe if any of these signals have a normal distribution.

A Q–Q plot, short for “quantile–quantile”, is typically used to assess whether a set of data potentially comes from a normal distribution. If the data are normally distributed, the points in the Q–Q plot will lie on a straight diagonal line. Conversely, the more the points in the plot deviate significantly from the straight diagonal line, the less likely the set of data follows a normal distribution. Figure 12 shows the Q–Q plot. Note that for all the variables, it appears there are normality violations across the data. Nevertheless, this is still not conclusive.

Therefore, we need to test this statistically to verify if the data are normally distributed. To this end, we used the Shapiro–Wilk test for normality. The null hypothesis for this test is that the data are normally distributed. The Prob < W value listed in the output is the *p*-value. If the chosen level is 0.05 and the *p*-value is less than 0.05, then the null hypothesis (i.e., the data are normally distributed) is rejected. If the *p*-value is greater than 0.05 the null hypothesis is not rejected. Table 9 shows the table with the results of the *p*-value for each type of observed measurement; it shows if the null hypothesis H0 is rejected or not. Finally, it shows the paired test to use. We observe that, for all measurements’ differences with and without sensors, if the *p*-value is larger than 0.05 (95% confidence), the normality is not rejected. Therefore, we must use *t*-test to obtain the mean difference between the two conditions and determine if the means of the monitored variables are different or not.

A paired *t*-test is used to test whether the means of the samples of both conditions are equal or not. Since a *t*-test requires us to know if the compared signals have the same variance, as a rule of thumb, we can assume that the populations have equal variances if the ratio of the larger sample variance to the smaller is less than 4:1. Since the variances for breathing frequency, heart rate, sleep rhythm, and body temperature sensors are 1.24269, 0.58489, 1.31356, and 1.30130, respectively, we consider that all measurements have equal variance. Table 10 shows the results of the paired *t*-test for all measurements. We observe that the means of the conditions on both scenarios, with and without sensors, are statistically different.

From the above analysis, we conclude that, in general, the subjects’ measurements are affected when subjects are aware of the presence of the system.

(B) Analysis by gender:

By plotting the box plot for the categories, we visualize associations between these two scenarios. Figure 13 shows box plots that allow us to visualize the type of association between “sensor” and “no-sensor” categories for men and women.

(C) Male subjects:

Figure 14 shows the histogram of the different measurements for male subjects. As shown, it is difficult to observe if any of this has a normal distribution. In the visual inspection it looks moderately normal and it is not conclusive. That is why it is to be complemented with a statistical test.

Figure 15 shows the Q–Q plot for all measurements of the male subjects. Note that for all the variables, there appears to be normality violations across the data. Nevertheless, this is still not conclusive.

Table 11 shows the results of the *p*-values for each type of measurement for the male subjects. From this table, we observe that three of the measurements can use the paired *t*-test and one requires the use of the Wilcoxon *t*-test.

The ratios of the larger to the smaller sample variance for breathing frequency, heart rate, sleep rhythm, and body temperature sensors for the male subjects are 1.11312, 0.34287, 2.32417, and 1.42631, respectively; therefore, we consider that all measurements have equal variance.

Table 12 shows the results of the paired *t*-test for all the measurements. Note that the means of the conditions on both scenarios, with and without sensors, are statistically different.

From the above analysis, we conclude that, in general, the “breathing frequency” and the “sleep rhythm” measurements for male subjects are affected when subjects are aware of the presence of the system.

(D) Female subjects:

Figure 16 shows the histogram of the difference of the different measurements for the female subjects. In this figure it is difficult to observe if any of this has a normal distribution.

Figure 17 shows the Q–Q plot for all measurements of the female subjects, and we observe that for all the variables, there appears to be divergences from normality in all data. However, this is not conclusive.

Table 13 shows the table with the results of the *p*-value for each type of measurement for male subjects. From this table, we observe that three of the measurements can use a paired *t*-test, and one requires the use of a Wilcoxon *t*-test. We observe that null hypothesis for “breathing frequency” is in the border to be rejected.

The ratio of the larger to the smaller sample variance for breathing frequency, heart rate, sleep rhythm, and body temperature sensors for males are 1.53213, 0.25152, 0.66405, and 1.15626 respectively; therefore, we consider all measurements to have equal variance.

Table 14 shows the results of the paired *t*-test for all measurements. We observe that, statistically, the means of the conditions in both scenarios, no-sensors and sensors, are different for “Breathing frequency”, “Heart rate”, “Sleep rhythm”, but not for “Body temperature”.

From the above analysis, we conclude that in general “Breathing frequency”, “Heart rate” and “Sleep rhythm” measures for female subjects are affected when they know they are under the system.

### 4.3. Algorithm Performance in Male and Female

Now, we analyze if the algorithm’s performance has any relationship with whether the subjects are men or women. The routines of older adults when they live alone can vary if we consider that a man lives alone or a woman lives alone. This can help us tailor the network and customize the sensors even more.

The tests run for 14 days, with the network running all the time. For seven days, the network works without the energy optimization algorithm implemented in the sensors. For the remaining seven days, we run the algorithm on the nodes. With the above, we can compare the batteries’ energy consumption and energy savings (ES). In Figure 18, we plot a close relationship represented in the energy saving for the consumption of the sensor network in each person of the chosen sample. We note that there is an additional 12% energy saving with the algorithm in houses of single women compared with houses of single men. This is interesting from the point of view of the person’s activity in the different areas of her house. Because the sensors represent a higher ranking for the active mode while the person uses some areas more than others, this generates greater consumption in the energy of the sensors. Interestingly, the lowest consumption is in the homes of older men, and this phenomenon may be because women are more active and move to a more significant number of areas in the house. This may lead to better adapting the algorithm for gender-based usage strategies, and could further help energy consumption.

### 4.4. Perceptions of Men and Women

In order to corroborate the perceptions of men and women, we conducted a survey based on a Likert scale with the following values: 0 = very bad, 1 = bad, 2 = OK, 3 = Good, 4 = Great. This survey was conducted every day of the week, and a general average was obtained. Figure 19 and Figure 20 show a radial map of perception, with concentric radii from 0 to 4, which are the previously mentioned sensation values of the scale. We observe that the days when people feel better with the presence of the network is from Monday to Friday and, specifically, on Monday, Tuesday, and Wednesday. In this aspect, both sexes coincide. Subsequently, women have a 10% greater sense of well-being than men. On weekends (Saturdays and Sundays), people feel good (or OK) with the care they could receive from the network. This may be because these are days when people receive more visits from family or friends, so people feel safer because they are not alone.

After a detailed analysis based on women and men separately to understand their degree of unconscious affectation, we will study the usefulness of the sensor network. Figure 17 describes the number of alerts by each sensor, due to an abnormality, per day of the week. The days of the week correspond to the colored boxes that each sensor has. The “x” axis has the labels of the types of sensors that make up the network. The “y” axis presents the number of alerts sent by each sensor on each day of the week. For example, on Sunday, the level sensor had 1 alert, the PTH sensor had 3 alerts, the motion sensor had 2 alerts, the noise sensor had 2 alerts, the gyroscope had 1 alert, the light and the air sensors did not present alerts. This graph gives an idea of the usefulness of the network for people. However, it is important to consider the false positives of the sensor alerts. In reality, an alert could have been the alarm of a possible accident or a cause that threatened the person’s health. With this figure, we observe the use of sensors by day of the week and its impact on sending alerts. We observe that the PTH sensor has the most significant impact due to the characteristics it measures. However, it only gives an idea of the general parameters of the home and that they could not necessarily put the person’s health at risk. The level sensor is the second sensor with the most significant impact on the network. It gives an idea of the person’s anomalous positions, which can indicate a fall. Light and air sensors contribute little to health care, but they improve people’s quality of life and comfort.

Although Figure 21 helps to know the impact of each sensor on the person’s health care, it is vitally important to know the true usefulness and degree of reliability of each sensor. The sensors can send alerts of possible attention situations, but these can be false positives, and the person is not necessarily in danger. Table 15 shows each sensor’s degree of confidence regarding the relationship between true alerts and alerts that were not at risk. It is essential to remember that not all sensors have the same degree of importance when detecting a possible danger. For example, the level sensor can be more accurate in detecting a possible fall of the person, and the motion sensor detects the presence of people in strange places at times that do not coincide with the person’s routine. This table only indicates the individual work of each sensor, but when we take the complete sensor network, this can be more useful for daily life. The light sensor does not apply to this degree of confidence because the luminaries are not directly related to the person’s health care. Therefore, this sensor only contributes to comfort and convenience. It is pertinent to consider that the network of this work is studied for the comfort, comfort, and health care of the older adult, without losing their independence. For this reason, all network sensors are essential to the extent that they contribute to people’s well-being.

### 4.5. Usefulness/Usability Metrics

This subsection is centered on the measurements of usefulness/usability by the user. These parameters are used in user-centered interaction design to evaluate a product through testing with the users themselves.

In designing any application or system, it is essential to reduce uncertainty and relying on quantifiable data obtained in research is advantageous. Therefore some metrics based on the user experience and the appreciation of the tool used to have this information are considered. These parameters are listed in the Table 16.

Table 17 shows the responses of each person who tested the sensor system in their home. They were asked the four questions in the Table 16 related to usability metrics for a system or application. The results show that 90% think that the monitoring system has Effectiveness. 92.5% think that the system has Efficiency. 95% are satisfied with the system. Finally, 25% feel that the system has the Learnability feature. This last result is curious and maybe because older people are not interested in learning elaborate technology-related aspects. Therefore, they may think that the system is complex. However, they did not need to do any installation or configuration.

## 5. Conclusions

Concerning society versus telecommunications technology, sensor networks create significant opportunities for the specialized lifestyle sector. This segment is experiencing rapid growth and change in its needs and can take advantage of technological innovations to offer more value and develop more products that generate value for consumers. From the home to commercial chains, they can improve the user experience thanks to a greater use of technology, which is efficiently and effectively achieved by the constant control and monitoring of people, preserving their independence and privacy, and taking care of their lifestyle.

The proposal’s objective is to improve the quality of life of older adults who live independently, preventing problems or deterioration in health. The key for technologies to penetrate the market for the elderly is to offer them attractive services with well-known functionality capable of arousing their interest, and getting manufacturers, promoters, and service providers to invest in this market. In other words, it is necessary to sell services, not technology, transmitting its benefits to the user at all times. We found that the impact of well-being is more significant with the presence of the network and that this also influences the person’s perception when they feel watched by the network. The four vital sign variables give a good insight into a person’s feelings of well-being and safety in different home areas. It is an excellent experiment to generate independence and confidence in older adults and propose a low-cost network to prevent accidents at home.

In this work, we performed experiments in homes for older adults who want to remain independent, using non-invasive technological monitoring. We have performed tests implementing a network of non-invasive sensors with measurement parameters such as level, luminosity, motion, pressure/humidity/temperature, air quality, and movement of doors and/or windows. For our experiments, we considered healthy people who do not suffer from cardiovascular or respiratory conditions. We measured four variables in people: heart and respiratory rates, body temperature, and sleep rhythm. In addition, we proposed a energy optimization algorithm network based on clustering, taking advantage of the person’s frequent location to prioritize sensors and take full advantage of the proactive and reactive nature of the packet routing protocol. We have found that people feel more comfortable and/or calm when the sensor network is active in their homes. We have classified our sample according to the appreciation of the network in men and women separately to analyze the network’s influence on their states of calm.

The statistical association between the monitored values for “heart rate” for men before and after installing sensors is slightly negative. Nevertheless, for females, the statistic association is a strong positive. The statistical association between the monitored values for “sleep rhythm” for both men and women before and after installing sensors is strongly negative.

Moreover, the statistical association between the monitored values for “body temperature” for men before and after installing sensors is positive. The same behavior is shown in females. Moreover, the association between these two variables is slightly stronger for men. We are also interested in knowing the effects of the measurements under these different conditions for male and female subjects.

## Figures and Tables

**Figure 1 ijerph-19-09966-f001:**
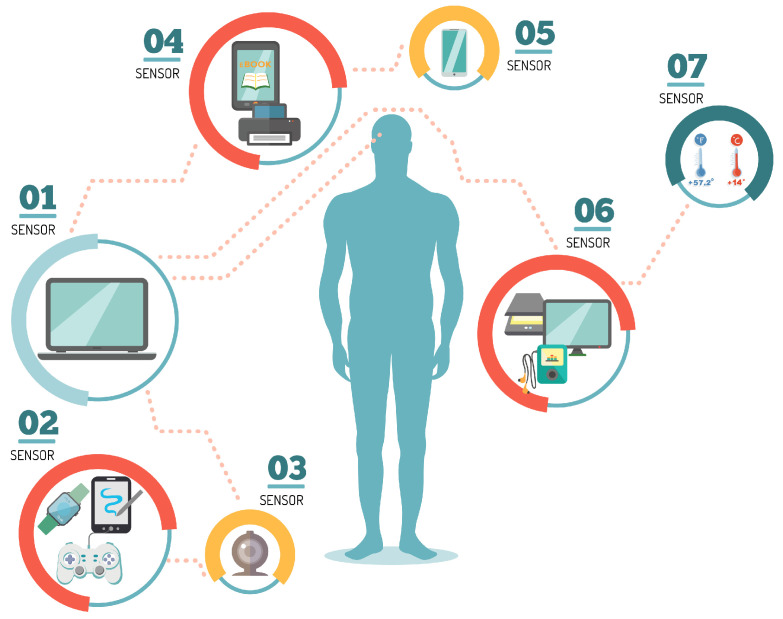
Types of body sensors for human wireless monitoring.

**Figure 2 ijerph-19-09966-f002:**
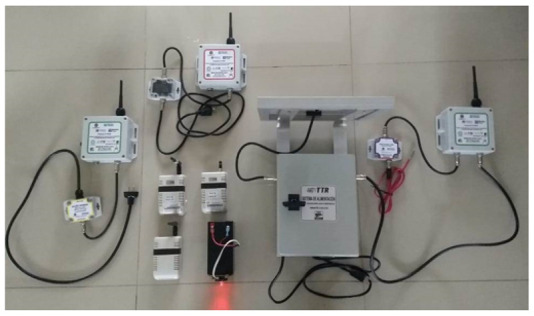
Wireless network with set of radio frequency nodes, coordinator node, and protective plastic coatings. Radio frequency node: country of origin: Mexico; brand: TTR; model: Node RF, µUSB polarization: 5 V Vcc at 100 mA; polarization RJ-11 RS-485 port: 5–12 V Vcc at 100 mA; operating frequency: UHF-2.4 GHz; connection port: RS-485, USB; material: ABS plastic; dimensions: 109.5 × 70 × 38 mm; white color. Radio frequency sensor/router: country of origin: Mexico; brand: TTR; model: Node RF; RS-485 port polarization: 5–12 V Vdc at 100 mA; end devices: 100; nodes in the network: 100; jumping capacity: 15; operating frequency: UHF-2.4 GHz; connection port: RS-485; material: ABS plastic; dimensions: 120 × 120 × 60 mm; white color.

**Figure 3 ijerph-19-09966-f003:**
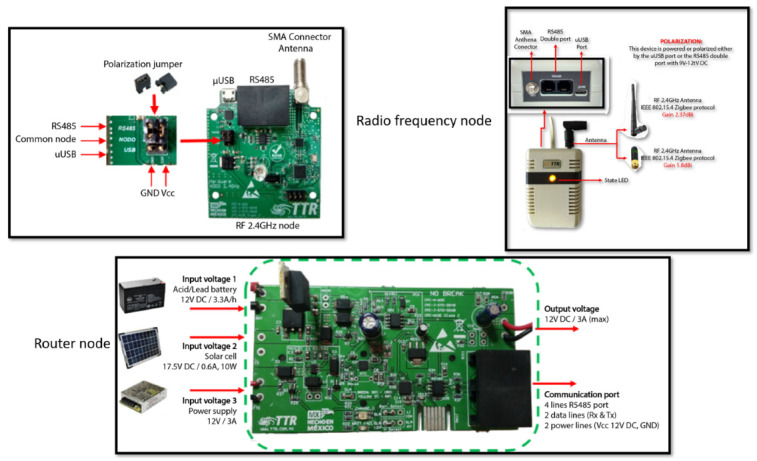
Router node and radio frequency nodes’ circuits, schematics, and connections.

**Figure 4 ijerph-19-09966-f004:**
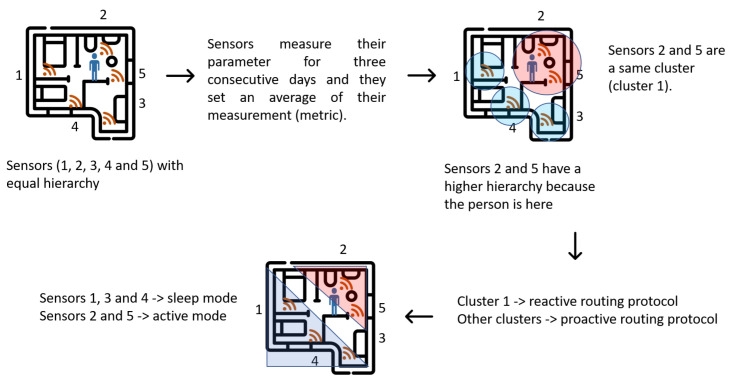
Sensor operation by hierarchies according to activity of person in house.

**Figure 5 ijerph-19-09966-f005:**
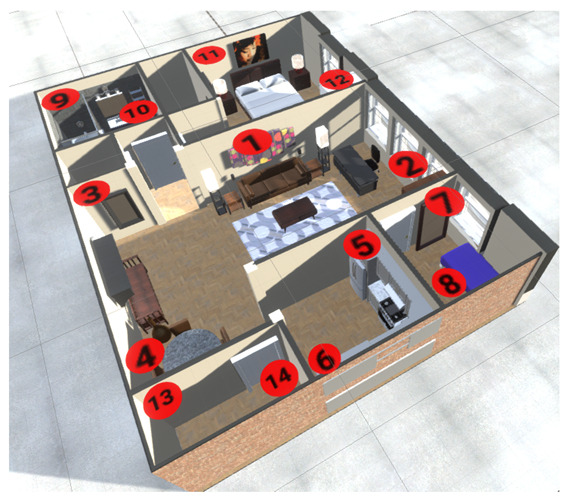
Diagram of sensor distribution by rooms in an average house for experimentation.

**Figure 6 ijerph-19-09966-f006:**
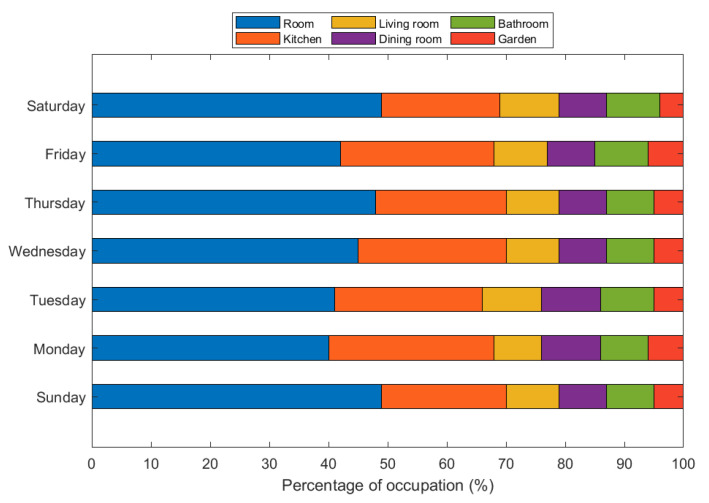
Average occupancy percentage per person in main areas of living place.

**Figure 7 ijerph-19-09966-f007:**
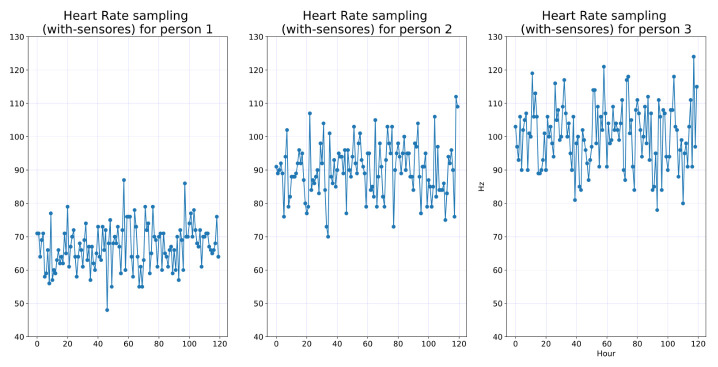
Time series of heart rate of three persons.

**Figure 8 ijerph-19-09966-f008:**
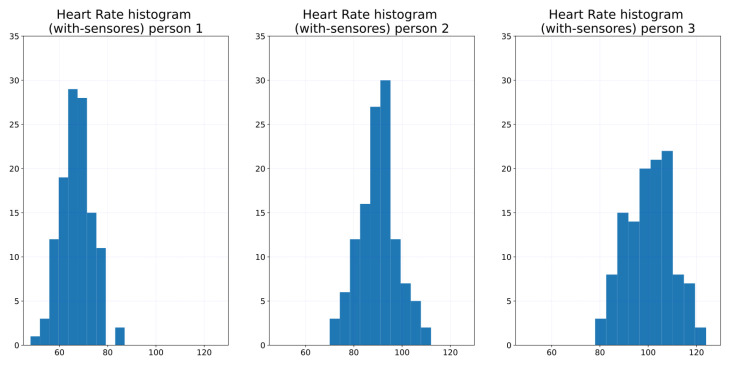
Heart rate histogram of three persons.

**Figure 9 ijerph-19-09966-f009:**
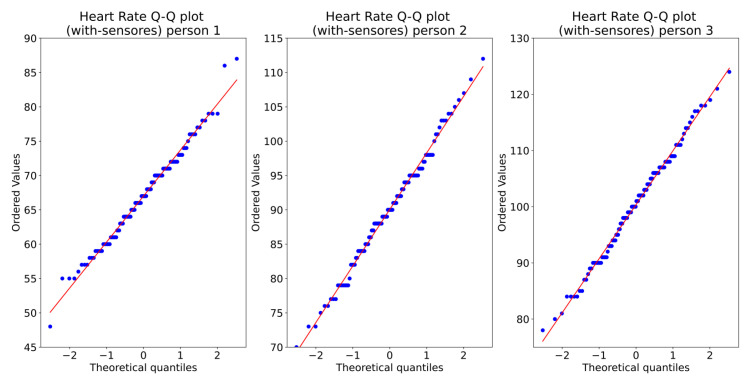
Heart Rate Q–Q plot of three persons.

**Figure 10 ijerph-19-09966-f010:**
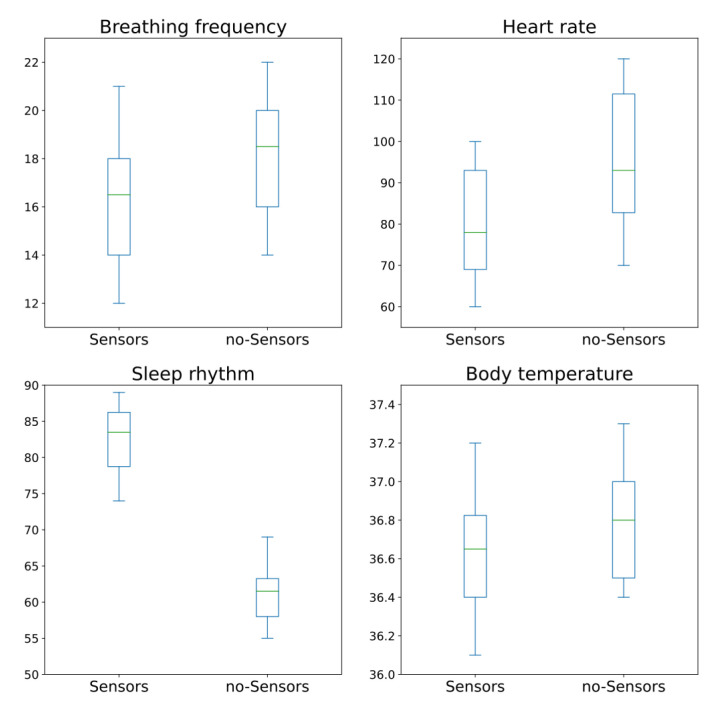
Box plots for “sensor” and “no-sensor” categories for all subjects.

**Figure 11 ijerph-19-09966-f011:**
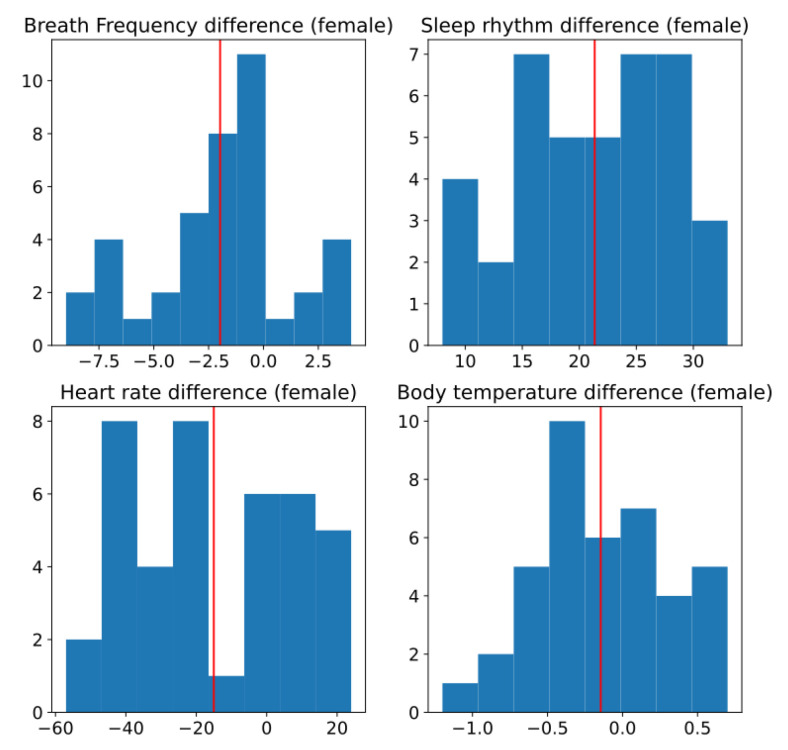
Histogram of difference between two conditions, without and with sensors. Red line is estimated mean.

**Figure 12 ijerph-19-09966-f012:**
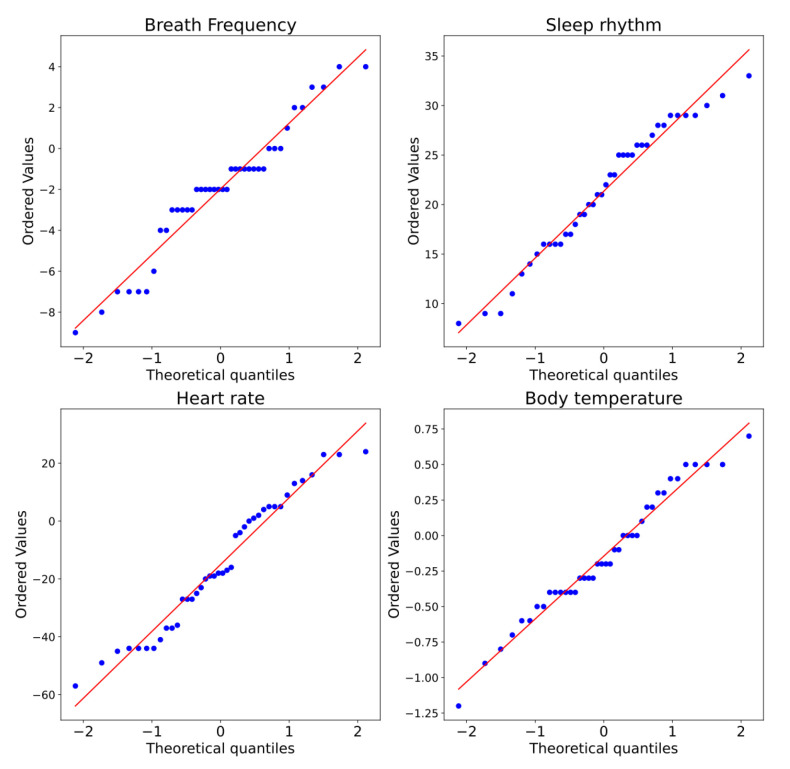
Q–Q plot of measurements captured for two conditions: without and with sensors.

**Figure 13 ijerph-19-09966-f013:**
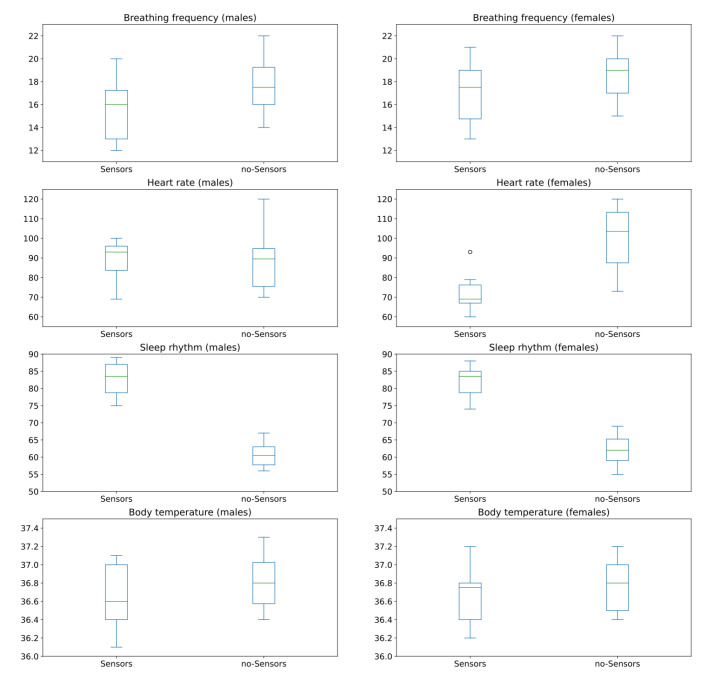
Box plots for sensor and no-sensor categories for men and women.

**Figure 14 ijerph-19-09966-f014:**
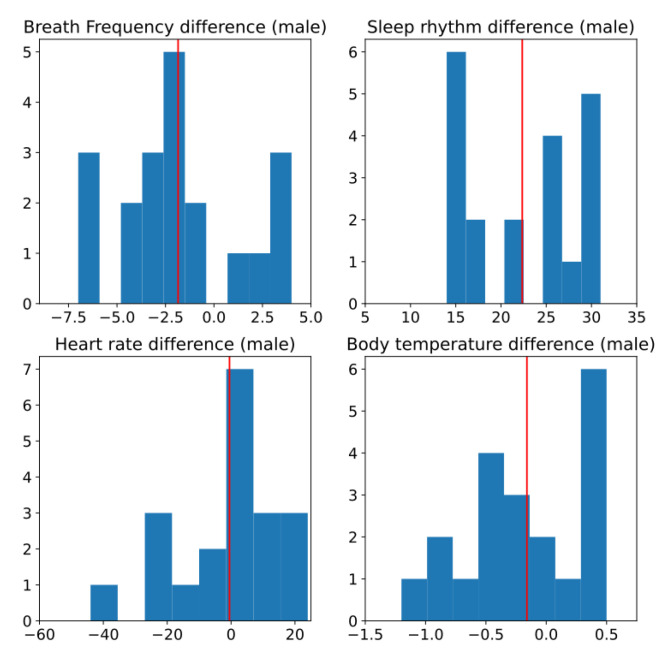
Histogram of difference between two conditions, without and with sensors, for male subjects. Red line is estimated mean.

**Figure 15 ijerph-19-09966-f015:**
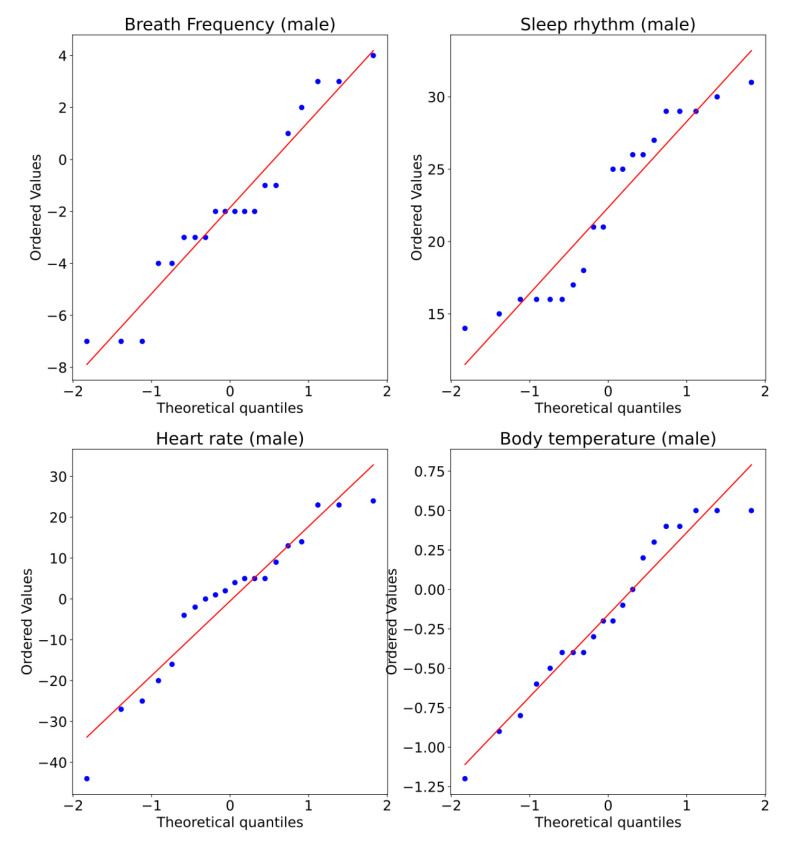
Q–Q plot of measurements captured for two conditions, without and with sensors, for male subjects.

**Figure 16 ijerph-19-09966-f016:**
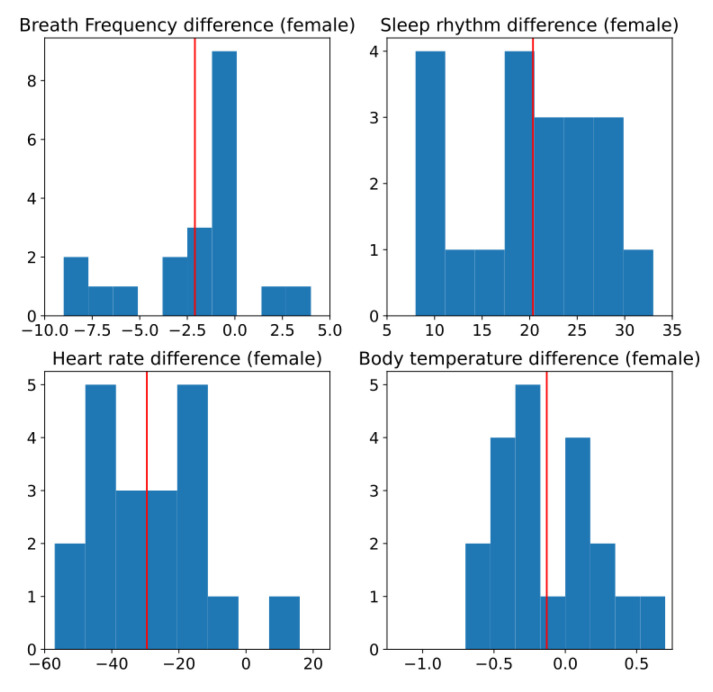
Histogram of difference between two conditions, without and with sensors, for female subjects. Red line is estimated mean.

**Figure 17 ijerph-19-09966-f017:**
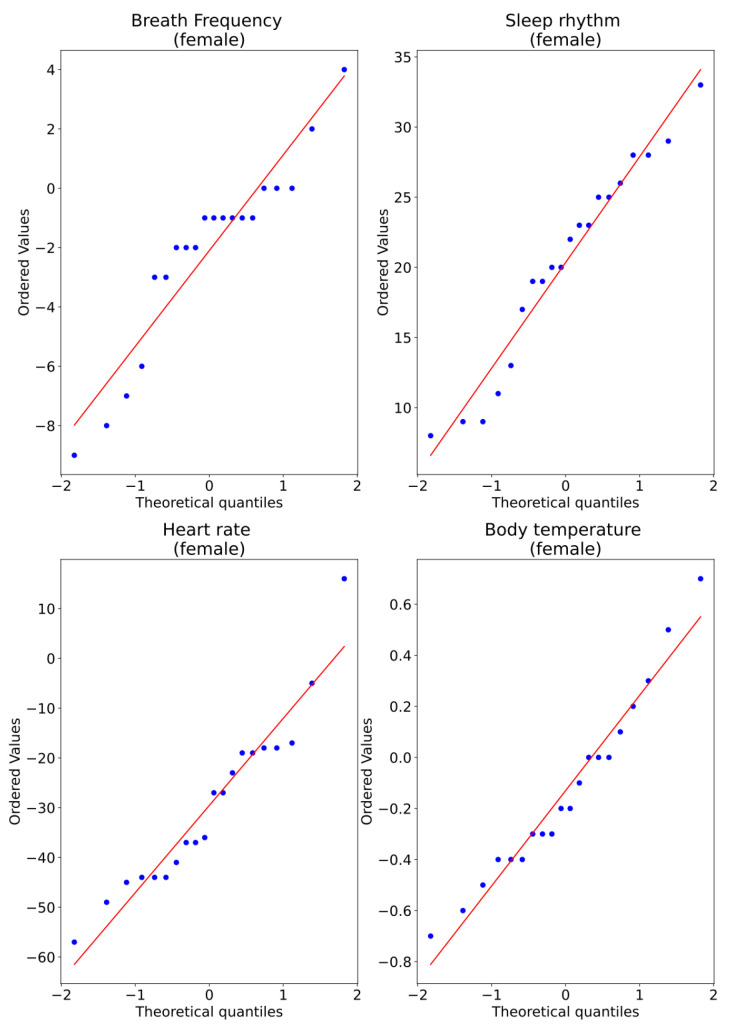
Q–Q plot of measurements captured for two conditions, without and with sensors, for female subjects.

**Figure 18 ijerph-19-09966-f018:**
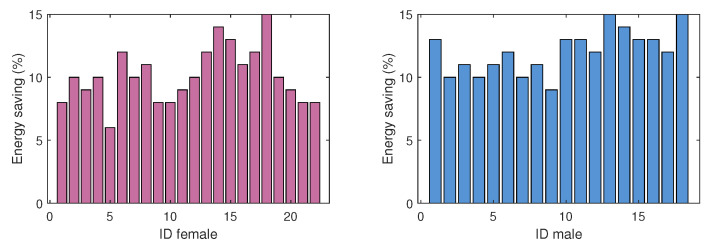
Energy saving (%) with algorithm in male- and female-occupied houses.

**Figure 19 ijerph-19-09966-f019:**
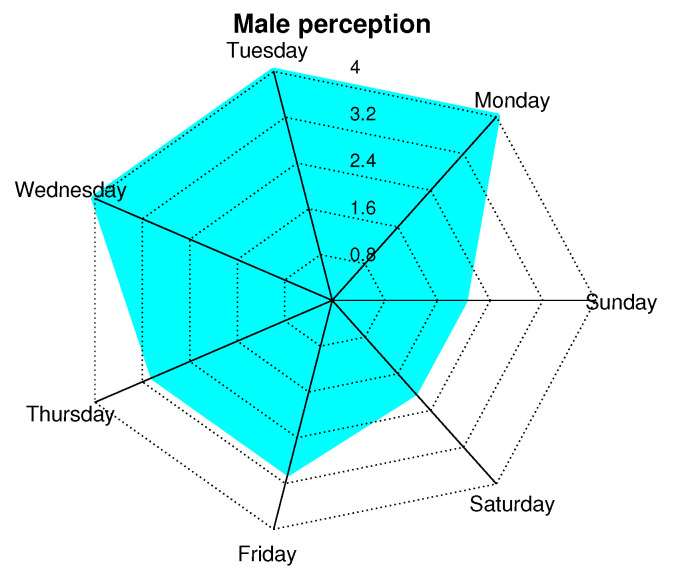
Perception of men about presence of network according to days of week.

**Figure 20 ijerph-19-09966-f020:**
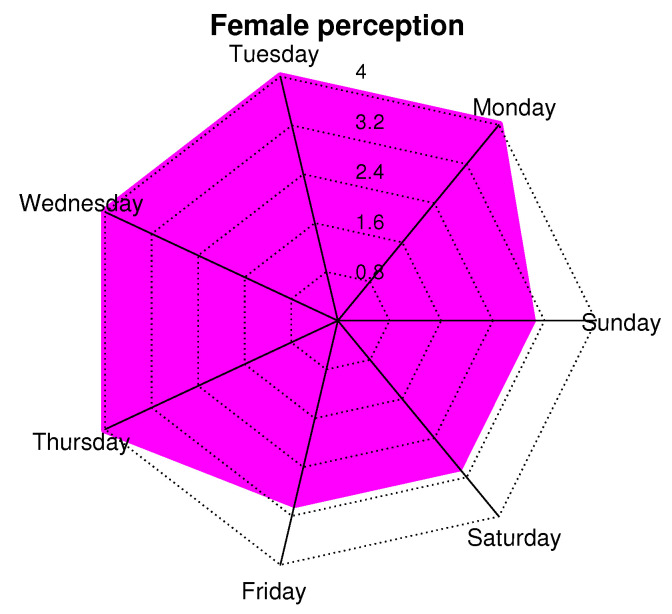
Perception of women about presence of network according to days of week.

**Figure 21 ijerph-19-09966-f021:**
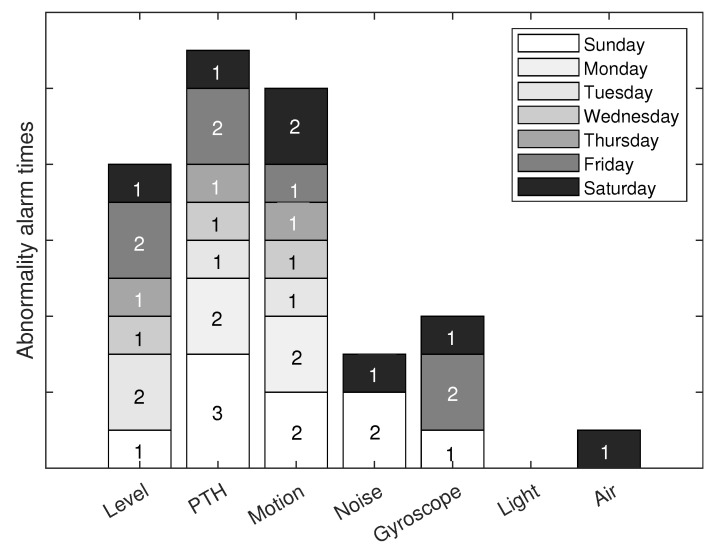
Number of alerts for any abnormality in house for each network sensor.

**Table 1 ijerph-19-09966-t001:** Core questions on cardiovascular and respiratory conditions.

Question
Are you diabetic?
Has any close relative (mother father, grandparents, or siblings) died suddenly before age 40 from a heart problem?
Do you have any heart diseases?
Do you have high blood pressure?
How often do you do physical exercise?
Have you felt dizzy when you exercise or play sports?
Have you ever had chest pain during exercise?
Have your nails or lips turned purple or bluish while exercising?
Have you felt that you cannot breathe after exercising?
Are you a smoker?
Do you have asthma?
Do you have exhaustion (tired) or lack of energy?
Does it present oxygenation above 90%?

**Table 2 ijerph-19-09966-t002:** Distribution of node devices in areas of house. These nodes can have one or more sensors connected to them.

ID	Sensor Type
1–2	Living room
3–4	Dining room
5–6	Kitchen
7–8	Room
9–10	Bathroom
11–12	Room
13–14	Garden

**Table 3 ijerph-19-09966-t003:** Connected sensor types in each zone of house. Each type of sensor measures a parameter of environment.

ID	Sensor Type
1	Level
2	PTH-Motion
3	Noise-Gyroscope
4	PTH
5	Level-Gyroscope
6	Motion
7	Motion-Gyroscope
8	Light
9	Level
10	Light-Gyroscope
11	PTH
12	Light-Motion
13	Air
14	PTH-Gyroscope

**Table 4 ijerph-19-09966-t004:** Change in person’s average heart rate per minute at each location in house.

Place of the House	Change in Heart Rate per Minute
Room	16% (↓)
Kitchen	12% (↓)
Living room	1% (↓)
Dining room	2% (↓)
Bathroom	18% (↓)
Garden	9% (↓)

**Table 5 ijerph-19-09966-t005:** Comparison of parameters and mechanisms in sensor networks for monitoring older adults at home.

Reference	Mechanism	Energy Optimization Algorithm	System Type	Estimation of Room Occupation	% Improvement of Perception in Men and Women	Experimental Validation Period
[21]	WiFi connection	Decision making devoted	Not reported	97%	Not reported	1 week
[40]	Observe, learn, and adapt	Adaptive learning system (clusters)	Not reported	Not reported	Not reported	3 weeks
[41]	Activities of daily living based on artificial intelligence	User-centered design sensors	Not reported	Not reported	Not reported	4 weeks
[42]	Wireless network 802.11 b/g	Ad hoc or infrastructure networks	Not reported	Not reported	Not reported	Not reported
[43]	Custom-designed RF ID system	Base-line tracking performance	Not reported	Not reported	Not reported	4 weeks
This work	WiFi connection	Clusters and hierarchies	Extended star	98%	8% for female and 5% for male	2 weeks

**Table 6 ijerph-19-09966-t006:** Remaining energy (RE) in each sensor with and without the application of algorithm.

ID Node	%RE without the Algorithm	%RE with the Algorithm
1	83%	92%
2	72%	86%
3	75%	85%
4	80%	91%
5	77%	88%
6	78%	86%
7	75%	86%
8	88%	94%
9	90%	96%
10	86%	94%
11	84%	90%
12	77%	88%
13	96%	98%
14	75%	84%

**Table 7 ijerph-19-09966-t007:** Heart rate 95% confidence intervals for each of these three persons.

95% Confidence Interval
[66.27625,67.95041667]
[88.83875,90.82416667]
[99.03395833,101.63125]

**Table 8 ijerph-19-09966-t008:** Heart rate mean estimators for each of these three persons.

Mean Estimators
67.00
90.06
100.36

**Table 9 ijerph-19-09966-t009:** Normality test and corresponding paired test to use.

Variables	*p*-Value for Shapiro–Wilk Normality Test	H0 = Samples Have Normal Distribution (with 95% Confidence)	Paired Test to Use
Breathing frequency	0.07835	H0 non-rejected	*t*-test
Heart rate	0.11033	H0 non-rejected	*t*-test
Sleep rhythm	0.2179	H0 non-rejected	*t*-test
Body temperature	0.54664	H0 non-rejected	*t*-test

**Table 10 ijerph-19-09966-t010:** Result of difference of means test.

Variables	*p*-Value for “Paired *t*-Test”	H0 = Samples Means Are the Same (with 95% Confidence)
Breathing frequency	0.000322	H0 rejected
Heart rate	$1.43\times 10^{-5}$	H0 non-rejected
Sleep rhythm	$1.91\times 10^{-36}$	H0 non-rejected
Body temperature	0.027598	H0 non-rejected

**Table 11 ijerph-19-09966-t011:** Normality test for male subjects and corresponding paired test to use.

Variables	*p*-Value for Shapiro–Wilk Normality Test	H0 = Samples Have Normal Distribution (with 95% Confidence)	Paired Test to Use
Breathing frequency	0.18171	H0 non-rejected	*t*-test
Heart rate	0.15011	H0 non-rejected	*t*-test
Sleep rhythm	0.02347	H0 rejected	Wilcoxon T
Body temperature	0.28179	H0 non-rejected	*t*-test

**Table 12 ijerph-19-09966-t012:** Result of difference of means test for male subjects.

Variables	Test	*p*-Value for “Paired *t*-Test”	H0 = Samples Means Are the Same (with 95% Confidence)
Breathing frequency	Paired *t*-test	0.015997866	H0 rejected
Heart rate	Paired *t*-test	0.90391866	H0 non-rejected
Sleep rhythm	Wilcoxon T	$8.65\times 10^{-5}$	H0 rejected
Body temperature	Paired *t*-test	0.112619049	H0 non-rejected

**Table 13 ijerph-19-09966-t013:** Normality test for male subjects and the corresponding paired test used.

Variables	*p*-Value for Shapiro–Wilk Normality Test	H0 = Samples Have Normal Distribution (with 95% Confidence)	Paired Test to Use
Breathing frequency	0.05744	H0 non-rejected	*t*-test
Heart rate	0.20114	H0 non-rejected	*t*-test
Sleep rhythm	0.36112	H0 non-rejected	*t*-test
Body temperature	0.55846	H0 non-rejected	*t*-test

**Table 14 ijerph-19-09966-t014:** Result of difference of means test for female subjects.

Variables	Test	*p*-Value for “Paired *t*-Test”	H0 = Samples Means Are the Same (with 95% Confidence)
Breathing frequency	Paired *t*-test	0.005909565	H0 rejected
Heart rate	Paired *t*-test	$2.3\times 10^{-9}$	H0 rejected
Sleep rhythm	*t*-test	$1.25\times 10^{-17}$	H0 rejected
Body temperature	Paired *t*-test	0.139209399	H0 non-rejected

**Table 15 ijerph-19-09966-t015:** Confidence percentage of real alerts from sensor network.

Sensor	Confidence
Level	88%
PTH	95%
Motion	77%
Noise	50%
Gyroscope	20%
Light	NA
Air	2%

**Table 16 ijerph-19-09966-t016:** Basic usability and user experience metrics.

Metric	Question
Effectiveness	Can users achieve their goal with the application or the system?
Efficiency	Is low mental effort required to launch the application or system?
Satisfaction	Can you reach your goal with minimal effort?
Learnability	Can you use the application or system without instructions in an intuitive way?

**Table 17 ijerph-19-09966-t017:** Older people who made use of sensor system in their homes.

ID Person	Effectiveness	Efficiency	Satisfaction	Learnability
1	✓	✓	✓	×
2	✓	✓	✓	×
3	✓	✓	✓	✓
4	✓	×	✓	×
5	✓	✓	✓	×
6	✓	✓	✓	×
7	✓	✓	✓	✓
8	✓	✓	✓	×
9	✓	✓	✓	×
10	×	✓	✓	×
11	✓	✓	×	✓
12	✓	✓	✓	×
13	✓	✓	✓	×
14	✓	✓	✓	×
15	✓	✓	✓	×
16	✓	×	✓	✓
17	✓	✓	✓	×
18	✓	✓	×	×
19	✓	✓	✓	×
20	✓	×	✓	×
21	✓	✓	✓	×
22	✓	✓	✓	×
23	✓	✓	✓	✓
24	✓	✓	✓	×
25	×	✓	✓	×
26	✓	✓	✓	×
27	✓	✓	✓	×
28	✓	✓	✓	×
29	✓	✓	✓	×
30	✓	✓	✓	×
31	×	✓	✓	×
32	✓	✓	✓	✓
33	✓	✓	✓	✓
34	✓	✓	✓	×
35	✓	✓	✓	✓
36	✓	✓	✓	×
37	✓	✓	✓	✓
38	✓	✓	✓	✓
39	✓	✓	✓	×
40	✓	✓	✓	×

## Data Availability

Data is contained within the article or Appendix A.

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
