# Peer review of "Statistical Study of User Perception of Smart Homes during Vital Signal Monitoring with an Energy-Saving Algorithm"

_ijerph, 2022, doi:10.3390/ijerph19169966_

Round 1

Reviewer 1 Report

Review of the Manuscript ID Number: ijerph-1818028 for the International Journal of Environmental Research and Public Health journal.

Title: Statistical study of user perception of smart homes during vital signal monitoring with an energy saving algorithm  

I would like to thank the authors and editors for having had the opportunity to review this manuscript.

The research regards the influence the sensor network used for optimizing home energy consumption based on people’s behavior on some of the inhabitants’ body parameters. The proposed optimization algorithm was based on sensors measuring pressure, temperature, humidity, motion, noise, gyroscope, light, and air in  main house spaces. The following parameters were monitored in participants: heart rate, respiratory rate, body temperature, and sleep rhythm.

Although authors referred to some relevant literature, the relation was general to a significant extent. They should try to extend the literature review by citing more studies specifically connected to the presented research.

I quite like the overall idea of this investigation, however I find it too general with a number of factors that were not included into the design and/or controlled. For instance, the measured parameters i.e.: heart rate, respiratory rate, body temperature, and sleep rhythm are easily affected by a number of issues such as sunny/cloudy level, some diseases, some physical problems related to performed work, bad mood, etc. These could significantly impact the obtained results and, thus, the provided inferences might not be appropriate.

The presented algorithm would be clearer if it was explained based on a more general diagram. It could also be interesting to know if other algorithms would perform better in the presented situation, or in different external weather conditions. One-week test seems to be too short to test the effectiveness of the proposed algorithm.

The measurement of the usefulness/usability could be much better related to some earlier studies in this area. Usually, these terms are examined in various dimensions by employing some validated and standardized questionnaires.

There is no comparison of the obtained results with similar studies of other investigators.

There are no figures from 7 to 14 ??

Given the above serious issues I do not recommend the paper to be published in its present form in the International Journal of Environmental Research and Public Health journal.  

Author Response

Reviewer 1 comments

Comment 1: I would like to thank the authors and editors for having had the opportunity to review this manuscript.

The research regards the influence the sensor network used for optimizing home energy consumption based on people’s behavior on some of the inhabitants’ body parameters. The proposed optimization algorithm was based on sensors measuring pressure, temperature, humidity, motion, noise, gyroscope, light, and air in main house spaces. The following parameters were monitored in participants: heart rate, respiratory rate, body temperature, and sleep rhythm.

Although authors referred to some relevant literature, the relation was general to a significant extent. They should try to extend the literature review by citing more studies specifically connected to the presented research.

Response: We thank to the Reviewer for his/her valuable and deep comments that will improve our work remarkably. The Reviewer is right, and we have substantially improved the Related Work subsection, including related work to the implementation of sensors for health monitoring. Also, we have included feature reviews and conditions for smart homes with medical technology applications. We have contrasted these approaches with the objective of our work.

The growing attention to energy efficiency in housing stimulates the expansion of "smart homes". A balance is required between energy efficiency and the needs of the occupants. The work cited in [32] addresses challenges faced by exploiting non-invasive wireless devices for user behavior monitoring. The application fields of intelligent energy management and elderly monitoring are chosen. It contrasts pretty well with our approach from the point of view of the occupation of the person detected non-invasively and energy savings in detection systems. As in the present work, the authors expose real cases in homes to analyze the network. Another work related to the previous one, from a more general perspective, is the one cited in [33]. The authors mention the ecological implication of smart metering, smart devices, and the smart home. From this approach, our work meets the Home Energy Management Systems requirements, as evidenced by the authors, such as minimizing energy demand and energy costs, Risk management, energy security, and Energy flexibility. Likewise, the work presented by the authors in [34] presents an international selection of the main smart home projects and the associated technologies of wearable/implantable monitoring systems and assistive robotics. This work reviews functions that can be implemented in smart houses and the equipment, objects, or appliances used. In this way, it complements our study by focusing our objective on classifying smart homes or systems with their equipment, fundamental algorithms, and functions used.

It is also essential to mention studies where the sensors are wearable by the user. In the work cited in [26], the authors comprehensively analyze telemedicine, which focuses on the client and server sides. They show that other studies associated with IoT-based smart home applications have several limitations that are not yet addressed, such as remote patient monitoring in home healthcare applications using IoT. In other literature review studies, such as the one cited in [35], the authors present a systematic review of the literature to determine levels of technology readiness among older adults and evidence of smart home-based information monitoring technologies for older adults. This last approach addresses an important aspect considered in our work. Thus, one of our objectives is to approach the prediction of disability and health-related quality of life non-invasive through a network of sensors. This problem can be extrapolated to remote medical care. For example, in work cited in [36], healthcare staff can also track the general health status of older people in real-time and provide feedback and support from distant facilities. This paper presents a comprehensive review of cutting-edge research and development in smart home-based remote healthcare technologies in this paper. This previous work is complemented by the paper cited in [37], where the authors provide an overview of the current state of intelligent health monitoring systems. They review these systems in intelligent environments from a general perspective and with particular attention to systems for the elderly and dependents. They also discuss the challenges for these systems from the perspective of technology development, system requirements, system design, and modeling.

Added references:

  1. Viani, F.; Robol, F.; Polo, A.; Rocca, P.; Oliveri, G.; Massa, A. Wireless architectures for heterogeneous sensing in smart home applications: Concepts and real implementation. Proceedings of the IEEE 2013, 101, 2381–2396.
  2. Fabi, V.; Spigliantini, G.; Corgnati, S.P. Insights on smart home concept and occu interaction with building controls. Energy Procedia 2017,111, 759–769.
  3. Chan, M.; Estève, D.; Escriba, C.; Campo, E. A review of smart state and future challenges. Computer methods and programs in biomedicine 2008, 91, 55–81.
  4. Liu, L.; Stroulia, E.; Nikolaidis, I.; Miguel-Cruz, A.; Rincon, A.R. Smart homes and home health monitoring technologies for older adults: A systematic review. International journal of medical informatics 2016, 91, 44–59. 665
  5. Majumder, S.; Aghayi, E.; Noferesti, M.; Memarzadeh-Tehran, H.; Mondal, T.; Pang, Z.; Deen, M.J. Smart homes for elderly health advances and research challenges. Sensors 2017, 17, 2496. 667
  6. Mshali, H.; Lemlouma, T.; Moloney, M.; Magoni, D. A survey on health monitoring systems for health smart homes. International Journal of Industrial Ergonomics 2018, 66, 26–56.

Comment 2: I quite like the overall idea of this investigation, however I find it too general with a number of factors that were not included into the design and/or controlled. For instance, the measured parameters i.e.: heart rate, respiratory rate, body temperature, and sleep rhythm are easily affected by a number of issues such as sunny/cloudy level, some diseases, some physical problems related to performed work, bad mood, etc. These could significantly impact the obtained results and, thus, the provided inferences might not be appropriate. 

Response:

Thank you very much. The reviewer is right in his concern, and we had forgotten to include a simple database that asked each person to be considered as part of the sample. We have included the initial experimentation questionnaire for classifying people and selected healthy people without heart or respiratory conditions. We add a table to clarify the initial conditions of investigation and to be able to outline, as far as possible, our experiment correctly. We have chosen healthy people who do sports at least three times a week and are non-smokers. Next, we present the set of measured characteristics and questions that were asked of older adults to be considered in this work.

Let us remember that the scope of this work is the monitoring of independent older adults who want to continue being so. Therefore, our study includes healthy people, as far as possible, in order to observe their slight changes in four variables related to vital signs.

Comment 3: The presented algorithm would be clearer if it was explained based on a more general diagram. It could also be interesting to know if other algorithms would perform better in the presented situation, or in different external weather conditions. One-week test seems to be too short to test the effectiveness of the proposed algorithm.

Response:

Thank you very much for your comment.

Figure [5] shows a general scheme of the operation of the proposed algorithm. Initially, the nodes are distributed throughout the house, and the person moves freely through it. All sensors have three days to have an average value of their measurement metric. When the person spends more time in the same place in the house, the sensors in that area increase their hierarchy and create a cluster. The other sensors in the house remain in their current hierarchy. Sensors with a higher hierarchy are put into active mode, and their routing protocol works reactively. While the other sensors are put into sleep mode for specific periods, and the routing protocol works in proactive mode.

Besides, we have added a subsection entitled "Degree of confidence of the sample within the period of experimentation".

Comment 4: The measurement of the usefulness/usability could be much better related to some earlier studies in this area. Usually, these terms are examined in various dimensions by employing some validated and standardized questionnaires.

Response:

We have added a subsection on measures of usefulness/usability by the user. These parameters are used in user-centered interaction design to evaluate a product through testing with users themselves.

This subsection is centered on measures of usefulness/usability by the user. These parameters are used in user-centered interaction design to evaluate a product through testing with users themselves.

In designing any application or system, it is essential to reduce uncertainty, and relying on quantifiable data obtained in research is advantageous. This is why some metrics based on the user experience and the appreciation of the tool used to have this information are considered. These parameters are listed in the Table [14].

Table [15] shows the responses of each person who tested the sensor system in their home. They were asked the four questions in the [14] Table related to usability metrics for a system or application. The results show that 90\% think that the monitoring system has Effectiveness. 92.5\% think that the system has Efficiency. 95\% are satisfied with the system. Finally, 25\% feel that the system has the Learnability feature. This last result is curious and maybe because older people are not interested in learning elaborate technology-related aspects. Therefore, they may think that the system is complex. However, they did not need to do any installation or configuration.

Comment 5: There is no comparison of the obtained results with similar studies of other investigators.

Response:

We have made a table of works related to the main parameters reported to contrast our work.

Table [5] reflects a search for works related to ours and shows a series of main analysis characteristics for this type of experiment. We report the type of mechanism used for caring for older adults at home. In addition to this mechanism, the wireless or wired system is proposed to track or locate the person. In addition, we report if there is an energy-saving technique to optimize network and home resources. Also, we observe the topology of the network of technological resources and if this care has a positive perception in people. Finally, we report the tests' execution time to validate the results.

Comment 6: There are no figures from 7 to 14 ??

Response:

We apologize to the Reviewer and the Journal because we had put an image format (.svg) that was incompatible with the MDPI template, and we did not realize that these figures did not appear. We have already added figures corresponding to the numbering from 7 to 14.

Reviewer 2 Report

This work highlights the importance of a network implemented in the home of elderly people with a simple home energy optimization. 

Technically the author describes the paper in a good way. 

However, these comments are to improve the presentation.

"Nowadays, intelligent design applied to the home sector has become an interesting approach to endow such spaces with safety and functional technology."

Need a reference!!

Line 52 needs a ref.

Line 77, And -->, and

There are too long paragraphs, for example, line 53 to 79,

You can divide it into 2~3 paragraphs, for example.

Can you describe the figures in labels more in detail?

Section 2.1 Variables:

We consider the variables as follows. First: Heart rate is ...

Blood font is not necessary.

Line 229, you can write in the label of the figure/picture.

Line 253 is the same.

Can you refer to the figures in your paragraphs before the figures themselves?

Line 264 needs more details.

Section 2.3. Sensor specifications

Would you make 1~2 lines why do you consider these parameters before the list?

You can use (A), (B), ...

Table 1, and Table 2 should be before Figure 5

Please explain in detail the tables in the labels

When you refer to the Table/Figure, please don't cut the idea. Instead, you can use it in one paragraph,

For example table 4 shows...., In addition, the table describe... Besides, the table concludes ...

Section 4.1,

Subsection: (A) testing Normality

Where are Figures 7, 8, 9, and others?

Line 444, 

use the numbers in 2 or 4~5 digits,

1.11311983471 -> 1.11312

Can you explain the conclusions more?

You can reduce other parts of the discussion and put in the conclusions, for example.

Usually, conclusions or introduction: 10% of total

Author Response

Reviewer 2 comments

Comment 1: This work highlights the importance of a network implemented in the home of elderly people with a simple home energy optimization.

Technically the author describes the paper in a good way.

However, these comments are to improve the presentation.

"Nowadays, intelligent design applied to the home sector has become an interesting approach to endow such spaces with safety and functional technology.”

Need a reference!!

Line 52 needs a ref.

Line 77, And -->, and

There are too long paragraphs, for example, line 53 to 79,

You can divide it into 2~3 paragraphs, for example.

Can you describe the figures in labels more in detail?

Section 2.1 Variables:

We consider the variables as follows. First: Heart rate is ...

Blood font is not necessary.

Line 229, you can write in the label of the figure/picture.

Line 253 is the same.

Can you refer to the figures in your paragraphs before the figures themselves?

Line 264 needs more details.

Section 2.3. Sensor specifications

Would you make 1~2 lines why do you consider these parameters before the list?

You can use (A), (B), ...

Table 1, and Table 2 should be before Figure 5

Please explain in detail the tables in the labels

When you refer to the Table/Figure, please don't cut the idea. Instead, you can use it in one paragraph,

For example table 4 shows...., In addition, the table describe... Besides, the table concludes ...

Section 4.1,

Subsection: (A) testing Normality

Where are Figures 7, 8, 9, and others?

Line 444,

use the numbers in 2 or 4~5 digits,

1.11311983471 -> 1.11312

Response: Thank you very much for your comment and your valuable contributions. We have improved typos extensively throughout the manuscript. We have complemented the labels of the figures with greater depth in their explanation. We have supplemented the information in Figure 3 to better describe the sensor circuitry. We have corrected the labels of the Figures and Tables and we have approximated the decimal numbers in the indicated paragraph.

We apologize to the Reviewer and the Journal because we had put an image format (.svg) that was incompatible with the MDPI template, and we did not realize that these figures did not appear. We have already added figures corresponding to the numbering from 7 to 14.

We have added the following references:

  1. Horváth, I. Connectors of smart design and smart systems. AI EDAM 2021, 35, 132–150.
  2. Tiersen, F.; Batey, P.; Harrison, M.J.; Naar, L.; Serban, A.I.; Daniels, S.J.; Calvo, R.A.; et al. Smart home sensing and monitoring in households with dementia: user-centered design approach. JMIR aging 2021, 4, e27047

Comment 2: Can you explain the conclusions more?

You can reduce other parts of the discussion and put in the conclusions, for example.

Usually, conclusions or introduction: 10% of total

Response: The Reviewer notes a very good point. We have improved the conclusion connecting it with our work contribution from beginning to end of the manuscript. We have highlighted this contribution in the conclusion section as well.

With this work, we have done experiments in homes for older adults who want to remain independent but with non-invasive technological monitoring. We have done tests implementing a network of non-invasive sensors with measurement parameters such as level, luminosity, motion, pressure/humidity/temperature, air quality, and turns of doors and/or windows. For our experiments, we consider healthy people who do not suffer from cardiovascular or respiratory conditions. We have measured four variables in people: Heart rate, Respiratory rate, Body temperature, and Sleep rhythm. In addition, we propose a network energy optimization algorithm based on clustering and taking advantage of the person's frequent location to prioritize sensors and take full advantage of the proactive and reactive nature of the packet routing protocol. We have found that people feel more comfortable and/or calm when the sensor network is active in their homes. We have classified our sample according to the appreciation of the network in men and women separately to analyze the network's influence on their state of calm.

The statistical association between the monitored values for "Heart rate" for men before and after installing sensors is slightly negative. Nevertheless, for females, the statistic association is a strong positive. The statistical association between the monitored values for "Sleep rhythm" for both men and women before and after installing sensors is strongly negative.

Moreover, the statistical association between the monitored values for "Body temperature" for men before and after installing sensors is positive. The same behavior is shown in females. Moreover, the association between these two variables is slightly stronger for men.

We also interested in knowing the effect of the measurements under these different conditions for male and female subjects.

Reviewer 3 Report

The manuscript is on a very interesting issue, analyzes a simple, easy-to-implement sensor network, optimized with an adaptive energy optimization algorithm to improve the lives of older in the home with the highest efficiency. The study proposes a network with different sensors, such as level, pressure/temperature/humidity, motion, noise, gyroscope, light, and air, to monitor the home’s main spaces. 

In the introduction the authors justify well the monitoring needed of the house space and inform us about the existing technology.

The material and methods are explained adequately and there are a lot of experiments with apparatus and procedures used.

In the results section, please explain them with discussion there are missing figures: 7-14, so the discussions are not complete.

Please elaborate consistent conclusions regarding added value not only future possible applications.

The references are appropriate.

Author Response

Reviewer 3 comments

Comment 1: The manuscript is on a very interesting issue, analyzes a simple, easy-to-implement sensor network, optimized with an adaptive energy optimization algorithm to improve the lives of older in the home with the highest efficiency. The study proposes a network with different sensors, such as level, pressure/temperature/humidity, motion, noise, gyroscope, light, and air, to monitor the home’s main spaces.

In the introduction the authors justify well the monitoring needed of the house space and inform us about the existing technology.

The material and methods are explained adequately and there are a lot of experiments with apparatus and procedures used.

Response: Thank you very much the Reviewer for their valuable comments and review of our work.

Comment 2: In the results section, please explain them with discussion there are missing figures: 7-14, so the discussions are not complete.

Response: The Reviewer is correct.

We apologize to the Reviewer and the Journal because we had put an image format (.svg) that was incompatible with the MDPI template, and we did not realize that these figures did not appear. We have already added figures corresponding to the numbering from 7 to 14.

Comment 3: Please elaborate consistent conclusions regarding added value not only future possible applications.

The references are appropriate.

Response: The Reviewer notes a very good point. We have improved the conclusion connecting it with our work contribution from beginning to end of the manuscript. We have highlighted this contribution in the conclusion section as well.

With this work, we have done experiments in homes for older adults who want to remain independent but with non-invasive technological monitoring. We have done tests implementing a network of non-invasive sensors with measurement parameters such as level, luminosity, motion, pressure/humidity/temperature, air quality, and turns of doors and/or windows. For our experiments, we consider healthy people who do not suffer from cardiovascular or respiratory conditions. We have measured four variables in people: Heart rate, Respiratory rate, Body temperature, and Sleep rhythm. In addition, we propose a network energy optimization algorithm based on clustering and taking advantage of the person's frequent location to prioritize sensors and take full advantage of the proactive and reactive nature of the packet routing protocol. We have found that people feel more comfortable and/or calm when the sensor network is active in their homes. We have classified our sample according to the appreciation of the network in men and women separately to analyze the network's influence on their state of calm.

The statistical association between the monitored values for "Heart rate" for men before and after installing sensors is slightly negative. Nevertheless, for females, the statistic association is a strong positive. The statistical association between the monitored values for "Sleep rhythm" for both men and women before and after installing sensors is strongly negative.

Moreover, the statistical association between the monitored values for "Body temperature" for men before and after installing sensors is positive. The same behavior is shown in females. Moreover, the association between these two variables is slightly stronger for men.

We also interested in knowing the effect of the measurements under these different conditions for male and female subjects.

Round 2

Reviewer 1 Report

Although the authors tried to improve the paper, I consider the main flaws (especially the experimental design with and the problems with the control of factors and the usability analysis) significant and the overall contribution as weak. Thus, I do not recommend the paper to be published in the International Journal of Environmental Research and Public Health journal. I also think that successfully improving the study design and analysis in this paper is unlikely as it would constitute an entirely different research.

Author Response

Dear

Editor

International Journal of Environmental Research and Public Health in the special issue “Advanced Rehabilitative and Assistive Engineering for the Elderly and People with Disabilities”.

We are submitting the paper:

“Statistical study of user perception of smart homes during vital signal monitoring with an energy saving algorithm”

Authored by: CAROLINA DEL-VALLE-SOTO*, JUAN ARTURO NOLAZCO-FLORES, JOSE ALBERTO DEL PUERTO-FLORES, RAMIRO VELÁZQUEZ, LEONARDO J. VALDIVIA, JULIO ROSAS-CARO, AND PAOLO VISCONTI.

We would like to thank the reviewers and editors for their detailed analysis of the manuscript; the comments are very valuable to us. In the revised version of the paper, we have incorporated the all changes recommended by the reviewers.

Comments to all observations and suggestions including point-by-point responses are addressed in the following text.

Reviewer 1 comments

Comment 1: Although the authors tried to improve the paper, I consider the main flaws (especially the experimental design with and the problems with the control of factors and the usability analysis) significant and the overall contribution as weak. Thus, I do not recommend the paper to be published in the International Journal of Environmental Research and Public Health journal. I also think that successfully improving the study design and analysis in this paper is unlikely as it would constitute an entirely different research.

Response: We thank to the Reviewer. We understand the Reviewer's concern and respect His/Her point of view. We have tried to improve a little more the formality of the methodology and experimentation.

Thank you very much.

Sincerely,

Carolina Del-Valle-Soto

Universidad Panamericana. Facultad de Ingeniería. Álvaro del Portillo 49, Zapopan, Jalisco, 45010, México.

Phone: +52 (33) 13682200 | Ext. 4866

Email: cvalle@up.edu.mx

Reviewer 2 Report

The 33 pages are too long, can you reduce it to 20 pages, if possible,

remove redundant data, and fix the figures, and tables!

Line 200 ~ Line 203 including Figure 1, please remove them, no need in this paper.

As I mentioned, labels should be described in two lines to make it clear for the audience.

Reduce the size. Use the white spaces of the figures as much as possible.

Line 163 to 196 should move to the part of the related work.

Section 2.1 Variables:

You can start your paragraph using First, Second, and so on.

For example: "We consider the variables as follows. First: Heart rate is ..."

Please use sequential numbers.

Line 248 is not clear.

The proposed system with its associated sensors 248 is shown in Figures 2 and 4.

Remove lines from 276 to 283 with its figures.

Line 308, remove "So, if,", >> "For instance, if the person uses some ....."

Summarize lines 315 to 328 in two lines.

Remove Algorithm 1, and replace it with figure 5

Remove lines 344 to 346

In figure 5, the label is not clear enough, add an explanation a little bit.

Each label should be around 15~25 words.

When you refer to the table/figure in the sentence, please use the table/figure after the reference.

Remove Tables 2 and 3. Other data, you can put in the Diagram, Figure 6!!

Figure 6, outside of the walls should be white color, not grey.

Figure 7, Y-axis should be percentage or rate. Move legend to the right side.

Remove table 5

Remove lines 394 to 399

Remove the references from the paper mentioned in table 5 if not necessary.

Change Table 6 to Column Chart.

Change Figure 8, and put the three persons in one figure, not three figures.

You can use color and a description in the figure itself.

Put Axis Title.

Remove Table 7. Instead, you describe it in sentences.

In figure 9, please put Axis Title, and modify it based on the standard.

Font, high, the width of the diagram.

Remove Figure 10, not necessary.

When you refer to the Figure, explain a little bit in sentences 1~3 lines and put the figure below directly.

Line 421, remove the link. The equation should be in inline and explain what is Xm, LE, n, x

Remove Table 8, and refer to the data in a sentence. Why important?

Line 424, remove it.

Line 433, remove >

Line 440, Testing Normality: >>> (A) Testing Normality

Line 414, and line 416... Figures????

Figure 11, the same, but three persons in one paragraph. Reverse axes.

Figure 12, put the four Figures horizontally not vertically.

Figure 13, is the same as Figure 12. Explain why Female?

Figure 14, is the same as Figure 13. Large the font based on the standard to be clear, and reduce the size of the Figures.

Line 460 to Line 470, please rewrite it statistically. Use the equation. Explain the symbols.

H0: ...

HA: ...

reduce the words, and explain table 9. Don't repeat the information.

Line 481,

(B) Analysis by Sex:

Analysis by Sex is not clear, please rewrite it again.

Figure 15 is not necessary. Otherwise, explain why it is crucial.

Figure 16 is the same as before.

Line 487, From this figure, remove these words and use "As shown, it is ..." and explain why.

Figure 17 is the same as before.

Figures 18, and 19, are the same.

Figure 20, change it to a line Chart.

Section 4.4 and 4.5 Remove it all and separate it as future work.

Please modify the conclusion and make it 10% of the whole paper.

Line 635, ravegu21??? 

References, use the important ones, reduce them.

Author Response

Dear

Editor

International Journal of Environmental Research and Public Health in the special issue “Advanced Rehabilitative and Assistive Engineering for the Elderly and People with Disabilities”.

We are submitting the paper:

“Statistical study of user perception of smart homes during vital signal monitoring with an energy saving algorithm”

Authored by: CAROLINA DEL-VALLE-SOTO*, JUAN ARTURO NOLAZCO-FLORES, JOSE ALBERTO DEL PUERTO-FLORES, RAMIRO VELÁZQUEZ, LEONARDO J. VALDIVIA, JULIO ROSAS-CARO, AND PAOLO VISCONTI.

We would like to thank the reviewers and editors for their detailed analysis of the manuscript; the comments are very valuable to us. In the revised version of the paper, we have incorporated the all changes recommended by the reviewers.

Comments to all observations and suggestions including point-by-point responses are addressed in the following text.

Reviewer 2 comments

Comment 1: The 33 pages are too long, can you reduce it to 20 pages, if possible, remove redundant data, and fix the figures, and tables!

Line 200 ~ Line 203 including Figure 1, please remove them, no need in this paper.

As I mentioned, labels should be described in two lines to make it clear for the audience.

Reduce the size. Use the white spaces of the figures as much as possible.

Line 163 to 196 should move to the part of the related work.

Section 2.1 Variables:

You can start your paragraph using First, Second, and so on.

For example: "We consider the variables as follows. First: Heart rate is ..."

Please use sequential numbers.

Line 248 is not clear.

The proposed system with its associated sensors 248 is shown in Figures 2 and 4.

Remove lines from 276 to 283 with its figures.

Line 308, remove "So, if,", >> "For instance, if the person uses some ....."

Summarize lines 315 to 328 in two lines.

Remove Algorithm 1, and replace it with figure 5

Remove lines 344 to 346

In figure 5, the label is not clear enough, add an explanation a little bit.

Each label should be around 15~25 words.

When you refer to the table/figure in the sentence, please use the table/figure after the reference.

Remove Tables 2 and 3. Other data, you can put in the Diagram, Figure 6!!

Figure 6, outside of the walls should be white color, not grey.

Figure 7, Y-axis should be percentage or rate. Move legend to the right side.

Remove table 5

Remove lines 394 to 399

Remove the references from the paper mentioned in table 5 if not necessary.

Change Table 6 to Column Chart.

Change Figure 8, and put the three persons in one figure, not three figures.

You can use color and a description in the figure itself.

Put Axis Title.

Remove Table 7. Instead, you describe it in sentences.

In figure 9, please put Axis Title, and modify it based on the standard.

Font, high, the width of the diagram.

Remove Figure 10, not necessary.

When you refer to the Figure, explain a little bit in sentences 1~3 lines and put the figure below directly.

Line 421, remove the link. The equation should be in inline and explain what is Xm, LE, n, x

Remove Table 8, and refer to the data in a sentence. Why important?

Line 424, remove it.

Line 433, remove >

Line 440, Testing Normality: >>> (A) Testing Normality

Line 414, and line 416... Figures????

Figure 11, the same, but three persons in one paragraph. Reverse axes.

Figure 12, put the four Figures horizontally not vertically.

Figure 13, is the same as Figure 12. Explain why Female?

Figure 14, is the same as Figure 13. Large the font based on the standard to be clear, and reduce the size of the Figures.

Line 460 to Line 470, please rewrite it statistically. Use the equation. Explain the symbols.

H0: ...

HA: ... 

reduce the words, and explain table 9. Don't repeat the information.

Line 481,

(B) Analysis by Sex:

Analysis by Sex is not clear, please rewrite it again.

Figure 15 is not necessary. Otherwise, explain why it is crucial.

Figure 16 is the same as before.

Line 487, From this figure, remove these words and use "As shown, it is ..." and explain why 

Figure 17 is the same as before.

Figures 18, and 19, are the same.

Figure 20, change it to a line Chart.

Section 4.4 and 4.5 Remove it all and separate it as future work.

Please modify the conclusion and make it 10% of the whole paper.

Line 635, ravegu21??? 

References, use the important ones, reduce them.

Response: Thank you very much to the Reviewer. Your comments have greatly improved the manuscript presentation.

We have fixed all the typos that the Reviewer mentions. We have improved the resolution, font, and distribution of the figures. We have made the main changes mentioned by the Reviewer.

In the part: “As shown, it is difficult to observe if any of this has a normal distribution”, thanks for the observation, and it is correct. Although in the visual inspection, it looks moderately normal, it is not conclusive. That is why it is to be complemented with a statistical test.

Table 5 was a specific request from another Reviewer.

Sections 4.4 and 4.5 are work that complements the appreciation and usability of the system. We do not consider it future work since it was implemented in this study.

Thank you very much.

Sincerely,

Carolina Del-Valle-Soto

Universidad Panamericana. Facultad de Ingeniería. Álvaro del Portillo 49, Zapopan, Jalisco, 45010, México.

Phone: +52 (33) 13682200 | Ext. 4866

Email: cvalle@up.edu.mx